# Continual GUI Agents

**Ziwei Liu**[1]  **Borui Kang**[2]  **Hangjie Yuan**[3]  **Zixiang Zhao**[4]  **Wei Li**[1]  **Yifan Zhu**[5]  **Tao Feng**[†6]

## Abstract

As digital environments (data distribution) are in flux, with new GUI data arriving over time—introducing new domains or resolutions—agents trained on static environments deteriorate in performance. In this work, we introduce Continual GUI Agents, a new task that requires GUI agents to perform continual learning under shifted domains and resolutions. We find existing methods fail to maintain stable grounding as GUI distributions shift over time, due to the diversity of UI interaction points and regions in fluxing scenarios. To address this, we introduce GUI-Anchoring in Flux (GUI-AiF), a new reinforcement fine-tuning framework that stabilizes continual learning through two novel rewards: Anchoring Point Reward in Flux (APR-iF 🖱️) and Anchoring Region Reward in Flux (ARR-iF 🔲). These rewards guide the agents to align with shifting interaction points and regions, mitigating the tendency of existing reward strategies to over-adapt to static grounding cues (e.g., fixed coordinates or element scales). Extensive experiments show GUI-AiF surpasses state-of-the-art baselines. Our work establishes the first continual learning framework for GUI agents, revealing the untapped potential of reinforcement fine-tuning for continual GUI Agents. The code is available at GUI-AiF 🔘

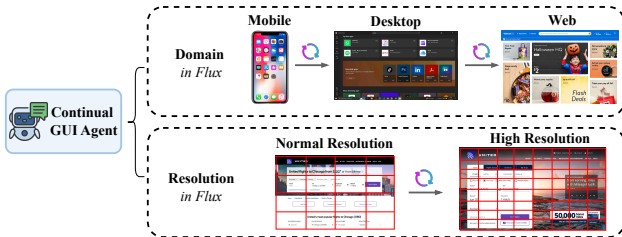

*Figure 1.* **Workflow.** Continual GUI Agents operate under two evolving scenarios: domain-in-flux (e.g., from Mobile OS to Web OS) and resolution-in-flux (e.g., scaling from 1080p to 4K).

structions, such as clicking icons or locating items within an interface (Hong et al., 2024; Guan et al., 2025). The core underlying this process is grounding, which refers to mapping textual instructions to precise pixel coordinates on the interactive interface (Zhao et al., 2025; Ye et al., 2026; Du et al., 2026). Currently, GUI grounding is typically achieved via Supervised Fine-Tuning (SFT) (Cheng et al., 2024; Gou et al., 2025; Lin et al., 2025; Wu et al., 2026) or Reinforcement Fine-Tuning (RFT) (Liu et al., 2025b; Gao et al., 2025; Yuan et al., 2026; Tang et al., 2026) on fixed datasets that encompass a variety of User Interface (UI) applications and interaction types (Cheng et al., 2024; Lin et al., 2025; Kapoor et al., 2024; Wu et al., 2025).

Current GUI agents are only trained from fixed UI datasets. However, digital environments in the real-world are in flux as new GUI data arrives over time. Consider the following examples: GUI agents may need to adapt to a new Operating System (OS) as updated versions are released, involving unfamiliar UI elements and layouts. Alternatively, GUI agents switch between OS platforms, e.g., from a Mobile OS (Cheng et al., 2024) to a Web OS (Lin et al., 2025). Even within the same OS, with device upgrades or customized OS versions for different devices, GUI agents are required to adapt not only to updated UIs but also to changes in screen resolution (e.g., 1080p to 4k (Li et al., 2025a)). To study this, we first propose two scenarios in Figure 1: i) continual learning across different UI domains (from mobile, desktop to web applications), ii) continual learning under changing resolutions (from normal to high resolution). The specific settings are illustrated in Sec. 4.1.

Subsequently, our in-depth analysis begins with the inherent dynamics of GUI environments. For instance, Mobile OS tends to include more textual elements, while Web OS pri-

## 1. Introduction

Autonomous GUI agents empower users to perform actions in various digital applications through natural language in-

---

[1]Department of Computer Science and Technology, Sichuan University, China [2]School of Computer Science, Nanjing University, China [3]College of Computer Science, Zhejiang University, China [4]Photogrammetry and Remote Sensing Lab, ETH Zürich, Switzerland [5]College of Computer Science, Beijing University of Posts and Telecommunications, China [6]Department of Computer Science and Technology, Tsinghua University, China. Correspondence to: Tao Feng[†] <fengtao.hi@gmail.com>.

*Proceedings of the 43rd International Conference on Machine Learning*, Seoul, South Korea. PMLR 306, 2026. Copyright 2026 by the author(s).

marily relies on icons, leading to significant differences in interaction points. Moreover, when the interface resolution varies from 1080p to 4K, significant proportional changes occur such as element scales (Li et al., 2025a). Under these dynamic conditions, the grounding capability of GUI agents becomes difficult to maintain, as reflected by their inability to consistently anchor interaction locations and element regions in flux. Recent advances in GUI grounding often follow either the SFT or RFT framework. Among these, SFT tends to memorize and fit the specific data distribution of the current task, making it inherently unsuitable for handling dynamic and diverse GUIs (Guo et al., 2025; Liu et al., 2025a; Shenfeld et al., 2025). In contrast, RFT leverages on-policy rewards to find solutions, and minimizes the KL divergence from the reference model. This creates an implicit gradient update to softly re-aligning parameters rather than disrupting them, naturally facilitating continual learning (Feng et al., 2022; Bian et al., 2024; Li et al., 2026; Feng et al., 2025; Lu et al., 2025; Liu et al., 2026) for GUI agents on new tasks (Lai et al., 2025; Jin et al., 2025; Shenfeld et al., 2025; Zhang et al., 2025b). Despite these advantages, the parameter-centric optimization of RFT primarily prevents drift from the reference model. When faced with domain or resolution shifts, UI interaction regions suffer from variations in both location and scaling (Shi et al., 2025; Lei et al., 2025), thus motivating a method to empower agents to continually learn in shifted UI environments.

In this paper, we introduce *GUI-AiF*, a Continual GUI Agents framework with the ability to perform *Anchoring in Flux*, which makes agents adapt to the flux of UI interactions. Specifically, the framework is designed to optimize policy within the RFT paradigm. Beyond the RFT advantages obtained from grounding policy, GUI-AiF incorporates a novel reward mechanism rooted in the principle of entropy maximization (Williams & Peng, 1991; Mnih et al., 2016). By keeping the entropy over predicted actions high, this mechanism encourages agents to anchor diverse interaction points and element regions, thereby mitigating grounding bias from static GUI tasks and enhancing adaptability for continual GUI migration. GUI-AiF introduces two rewards: **A**nchoring **P**oint **R**eward **i**n **F**lux (**APR-iF** 🔭) and **A**nchoring **R**egion **R**eward **i**n **F**lux (**ARR-iF** ⤵). APR-iF enhances the agents' ability to generalize interaction points by rewarding exploration of diverse locations, avoiding over-adapting to coordinates from a static task. ARR-iF promotes robustness to scale variations by encouraging diversity in predicted element regions, helping the agents adapt to different sizes across GUI tasks. In summary, the main contributions are following:

- We introduce *Continual GUI Agents*, and establish two novel scenarios (shifting domains and resolutions) for GUI agents to perform continual learning.

- We propose *GUI-AiF* to optimize the grounding policy reward within the RFT paradigm via **APR-iF** (🔭) and **ARR-iF** (⤵) modules, mitigating policy over-adaption on a single task and enhancing agents' continual ability in shifted UI tasks.

- Our GUI-AiF achieves state-of-the-art performance on the ScreenSpot-V1, V2, and Pro benchmarks, outperforming existing baselines.

## 2. Related Works

### 2.1. GUI Agents

GUI agents understand graphical user interfaces through natural language instructions, enabling automated execution of digital human-computer interaction tasks (Hong et al., 2024; Cheng et al., 2024; Lin et al., 2025; Yuan et al., 2026; Gao et al., 2025; Tang et al., 2026; Wang et al., 2024; Gou et al., 2025; Qin et al., 2025; Zhang et al., 2025a; Liu et al., 2025b). Early GUI approaches rely on structured representations like HTML code (Wen et al., 2024), and designed workflow modules such as planners (Wang et al., 2024) on closed-source large language models (Achiam et al., 2023; Li et al., 2025b). By fine-tuning models on GUI corpora, agents are aligned with human-like processing through vision-language interaction, leading to significantly improved grounding ability across various GUI contexts (Gou et al., 2025; Tang et al., 2025; Wu et al., 2025; Tang et al., 2026). As one of the fine-tuning paradigms, SFT is the pioneering method leveraged to build the GUI grounding (Cheng et al., 2024; Gou et al., 2025; Qin et al., 2025; Lin et al., 2025; Wu et al., 2026), while the dependency on extensive labeled datasets curtails its adaptability, leading to generalization challenges when faced with unseen scenarios. RFT leverages policy rewards to guide GUI grounding behavior (Tang et al., 2026; Yuan et al., 2026; Gao et al., 2025; Lu et al., 2026; Luo et al., 2025; Liu et al., 2025b), hence significantly enhancing performance. However, current GUI agents overlook the flux as new data appears, such as varying OS platforms and interface resolutions, hence motivating our GUI-AiF for Continual GUI Agents.

### 2.2. Continual Post-Training

Post-training shares similar foundations with Continual GUI Agents. SFT paradigm often struggles as it leverages cross-entropy objective forces the model to match the specific label distribution of the current task, leading to its convergence being arbitrarily far from previously learned capability (Shenfeld et al., 2025; Guo et al., 2025; Liu et al., 2025a). On the other hand, RFT offers potential advantages as its on-policy updates inherently favor solutions with minimal KL divergence from the base model, which is strongly correlated with reduced forgetting (Shenfeld et al., 2025; Jin

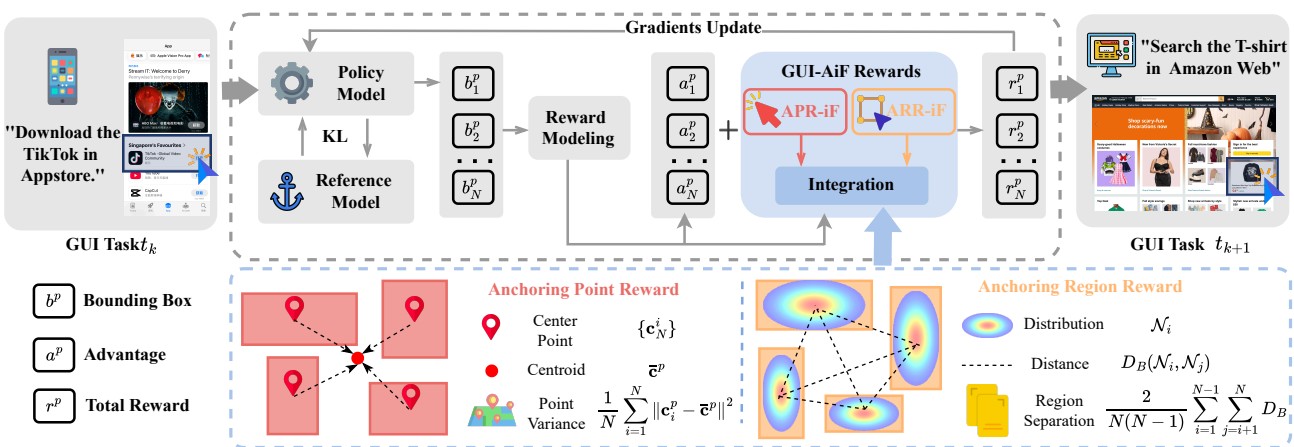

*Figure 2.* **Overview.** The Above shows the Continual GUI Agents pipeline. GUI-AiF is proposed to advance the RFT paradigm in this setting by shaping grounding rewards: APR-iF (🖱) encourages diverse interaction points, while ARR-iF (⬛) refines element regions. Together, GUI-AiF enables agents to better adapt to varying interaction locations and element scales.

et al., 2025; Liu et al., 2025c; Zhang et al., 2025b). Additionally, RFT's reward mechanism is able to implicitly scale parameter updates, hence alleviating the forgetting of prior knowledge (Jin et al., 2025). The efficacy of rule-based reward reinforcement learning, notably demonstrated by Group Relative Policy Optimization (GRPO) (Shao et al., 2024), which has been successfully applied across various domains (Shen et al., 2023; Li et al., 2025c), makes its use within the RFT framework. However, standard RFT optimization, including GRPO advantage estimation and KL divergence, is designed towards maximizing performance on the current task distribution. It lacks adaptability when encountering sequences with drastic shifts in visual layout, interaction domain, and resolutions inherent to GUI tasks.

## 3. Method

We introduce GUI-AiF for the proposed Continual GUI Agents task, centering the grounding optimization by integrating rewards of predicted interaction locations and element scales into RFT policy. As illustrate in Figure 2, GUI-AiF comprises two key innovations: APR-iF (🖱) rewards exploration of interaction points; ARR-iF (⬛) encourages diversity in predicted element regions, to enhance the adaptability of agents against the flux of GUI environments.

### 3.1. Problem Formation

The fundamental task of GUI grounding is to align a user's natural language instruction with its corresponding interactive element on a graphical interface at the pixel level. Given a interface $s$ and an instruction $i$, the model predict a bounding box by a fixed prompt $\mathbf{b}^p = [x_1^p, y_1^p, x_2^p, y_2^p]$ that localizes the region required by $i$, where $(x_1, y_1)$ and $(x_2, y_2)$ denote the top-left and bottom-right corners respectively. We also assume that the continual GUI tasks

as $\mathcal{T}_D$ or $\mathcal{T}_R$ and tasks arrive sequentially, where $\mathcal{T}_D = \{t_{d1}, t_{d2}, ..., t_{dn}\}$ represents tasks from distinct domains and $\mathcal{T}_R = \{t_{r1}, t_{r2}, ..., t_{rn}\}$ denotes tasks with shifting interface resolutions. Learning across such sequences requires the agents to continually adapt to the GUI task flux, preventing over-adaptation to any single task $t_k$.

In standard RFT policy, the model generates a sequence of tokens representing the predicted bounding box $\mathbf{b}^p$, encompassing both the interaction location (center point) and element scale (predicted region). While the reward policy method computes advantages solely by comparing predictions $\{\mathbf{b}_i^p\}$ to the ground truth $\{\mathbf{b}_i^{gt}\}$, this focuses on optimizing the *current* task, tending to over-adapt on coordinates and scales at specific GUI layouts. Our GUI-AiF modifies this objective by integrating two rewards designed to promote grounding exploration, achieved by encouraging both varying predicted center points and element regions.

### 3.2. GUI-AiF Rewards

**Anchoring Point Reward in Flux**. To enhance the agents' capacity to generalize interaction locations, avoiding over-adapting in particular coordinates of a learned single task, this APR-iF (🖱) encourages the exploration of diverse interaction points of predicted bounding boxes $\mathbf{b}^p$. Given a set of $N$ predicted $\{\mathbf{b}_1^p, \mathbf{b}_2^p, ..., \mathbf{b}_N^p\}$ generated by the agents for an instruction, APR-iF (🖱) first computes their corresponding center points $\{\mathbf{c}_1^p, \mathbf{c}_2^p, ..., \mathbf{c}_N^p\}$ where each center point $\mathbf{c}_i^p$ is calculated from its bounding box $\{\mathbf{b}_i^p\} = [x_{1,i}^p, y_{1,i}^p, x_{2,i}^p, y_{2,i}^p]$ as $c_i^p = \left(\frac{x_{1,i}^p + x_{2,i}^p}{2}, \frac{y_{1,i}^p + y_{2,i}^p}{2}\right)$. Physically, each $\mathbf{c}_i^p$ defines a candidate anchor point proposed by the agents based on the instruction understanding. To quantify the diversity among these proposed points, APR-iF (🖱) computes their spatial variance $\mathcal{R}_v$. It first calculates the centroid $\bar{c}^p = \frac{1}{N} \sum_{j=1}^{N} c_j^p$, which represents the aver-

age interaction point, then $\mathcal{R}_p$ is formulated as the average squared Euclidean distance between each individual point $\mathbf{c}_i^p$ and average prediction $\bar{c}^p$:

$$\mathcal{R}_p = \frac{1}{N} \sum_{i=1}^{N} |c_i^p - \bar{c}^p|^2 . \tag{1}$$

Intuitively, a higher variance indicates that center points are more spread out across the interface, rather than being clustered around a single coordinate. It helps the agents adapt to different interaction locations in GUI environments.

**Anchoring Region Reward in Flux**. Beyond diversifying interaction points, continual GUI grounding should also adapt to varying element scales, reflected in the predicted element regions. The ARR-iF (⬒) addresses this by rewarding spatial separation among the predicted bounding boxes. ARR-iF (⬒) first models each predicted bounding box $\mathbf{b}_i^p$ as a Gaussian distribution $\mathcal{N}_i(\mu_i, \Sigma_i)$, which transforms the predicted region into a continuous probability distribution, allowing us to formally measure their separation. The mean $\mu_i$ of each distribution is its center $\mathbf{c}_i^p$, and the covariance matrix $\Sigma_i$ is a diagonal matrix whose variances are proportional to the bounding box's width and height, which encodes the predicted scale and shape of the element region. To quantify the separation between any two predicted regions, ARR-iF (⬒) computes the Bhattacharyya distance $D_B$ between them. For two probability distributions $\mathcal{N}_i$ and $\mathcal{N}_j$, this distance is derived from their Bhattacharyya coefficient and has a closed-form solution which can be formulated as:

$$D_B(\mathcal{N}_i, \mathcal{N}_j) = \frac{1}{8}(\mu_i - \mu_j)^T \Sigma_{avg}^{-1}(\mu_i - \mu_j)$$
$$+ \frac{1}{2} \ln\left(\frac{\det(\Sigma_{avg})}{\sqrt{\det(\Sigma_i)\det(\Sigma_j)}}\right), \tag{2}$$

where $\Sigma_{avg} = \frac{\Sigma_i + \Sigma_j}{2}$ is the average covariance. Finally, the total regions separation $\mathcal{R}_r$ is defined as the average $D_B$ over pairs from all $N$ predictions:

$$\mathcal{R}_r = \frac{2}{N(N-1)} \sum_{i=1}^{N-1} \sum_{j=i+1}^{N} D_B(\mathcal{N}_i, \mathcal{N}_j). \tag{3}$$

This reward $\mathcal{R}_s$ incentivizes the agents to generate spatially distinct prediction regions, hence handling the diverse element scales encountered across fluxional GUI tasks.

**Integration Rewards**. We formulate APR-iF (🖱) and ARR-iF (⬒) rewards to $\mathcal{R}_{AiF}$ as a weighted sum, to consider the generalization of interaction locations and element scales:

$$\mathcal{R}_{AiF} = \alpha \cdot \mathcal{R}_p + \gamma \cdot \mathcal{R}_r, \tag{4}$$

where $\alpha$ and $\gamma$ are hyperparameters for each reward, which control the intensity of exploration over diverse points and

*Table 1.* Illustration of typical and continual GUI agents.

| Definition | Description | Scenario |
|---|---|---|
| **Typical GUI Agents** | Joint training on aggregated datasets | Static UI environment |
| **Continual GUI Agents** | Continual training on sequential datasets | 1) Mobile → Desktop → Web 2) Normal → High Resolution |

regions. By integrating $\mathcal{R}_{AiF}$ into the RFT policy, GUI-AiF strikes a trade-off between optimizing for the current task and promoting adaptability for future unseen tasks.

### 3.3. Reinforcement Fine-tuning with GUI-AiF

The RFT employs a policy method to compute a score $r_i$ for each prediction by comparing it against the ground truth, and obtains a set of reward scores $\{r_1, r_2, ..., r_N\}$. We leverage the GRPO method to calculate the relative policy advantage (Shao et al., 2024). Assuming $A_i$ is the $i$-th prediction, GRPO normalizes its score against the mean and standard deviation of the entire group of scores:

$$A_i = \frac{r_i - Mean(\{r_1, r_2, ..., r_N\})}{Std(\{r_1, r_2, ..., r_N\})}. \tag{5}$$

To control the parameters drift from the reference model, a KL divergence $\mathbb{D}_{KL}$ is further incorporated into the objective function. This term regularizes the policy updates by measuring the divergence between the current policy $\pi_\theta$, and a reference policy $\pi_{ref}$, which is typically the model from before the optimization step.

Functionally, $\mathcal{R}_{AiF}$ serves as a reward signal designed to encourage agents to explore interaction points and element regions that are in shifted location and scaling. The standard GRPO advantage $A_t$ computes correctness compared with ground-truth, driving the policy to maximize performance on the current task. However, such exploitation risks overfit to current UI layouts. To mitigate this, $\mathcal{R}_{AiF}$ is introduced as a counterbalance to the exploitation tendency, explicitly reshaping the optimization landscape to favor policies that are not only accurate but also generalized robust, thereby preserving the agent's continual learning ability for future environmental flux.

Therefore, we combine $A_i$ with the $\mathcal{R}_{AiF}$ as a comprehensive advantage at a given timestep $t$, and form $\mathcal{J}(\theta)$ as the objective optimization to guide the GUI agents' updates. The $\mathcal{J}(\theta)$ can be formulated:

$$\mathcal{J}(\theta) = \mathbb{E}_{\tau \sim \pi_\theta} [r_t(\theta)(A_t + \mathcal{R}_{AiF})$$
$$- \beta \cdot \mathbb{D}_{KL}[\pi_{ref}(\cdot|s_t)||\pi_\theta(\cdot|s_t)]], \tag{6}$$

where $r_t(\theta) = \frac{\pi_\theta(a_t|s_t)}{\pi_{ref}(a_t|s_t)}$ is the probability ratio of the current and reference policies, $\beta$ is a hyperparameter that controls the strength of the KL divergence term.

*Table 2.* Performance (%) in continual GUI domain on ScreenSpot-V1 and ScreenSpot-V2. The M., D., and W. represent the Mobile, Desktop, and Web platforms. The upper bound comes from the optimal results of typical GUI Agent method and serves as a reference.

| Method | | SSv1 Accuracy (%) | | | | | | | SSv2 Accuracy (%) | | | | | | |
|---|---|---|---|---|---|---|---|---|---|---|---|---|---|---|---|
| | | Mobile | | Desktop | | Web | | Avg. | Mobile | | Desktop | | Web | | Avg. |
| | | Text | Icon | Text | Icon | Text | Icon | | Text | Icon | Text | Icon | Text | Icon | |
| *Proprietary Model* | | | | | | | | | | | | | | | |
| GPT-4o (OpenAI, 2024) | | - | - | - | - | - | - | 18.8 | - | - | - | - | - | - | 20.1 |
| *Open-source Model* | | | | | | | | | | | | | | | |
| Qwen2.5VL-3B (Bai et al., 2025) | | 90.2 | 72.9 | 78.4 | 57.1 | 80.9 | 64.1 | 72.1 | 88.5 | 74.4 | 78.9 | 58.6 | 76.9 | 64.5 | 73.6 |
| *SFT-based GUI Method* | | | | | | | | | | | | | | | |
| SeeClick (Cheng et al., 2024) | Upper Bound | 78.0 | 52.0 | 72.2 | 30.0 | 55.7 | 32.5 | 53.4 | 78.4 | 50.7 | 70.1 | 29.3 | 55.2 | 32.5 | 55.1 |
| | M. | 52.7 | 59.8 | 52.1 | 42.9 | 20.4 | 25.7 | 42.3 | 51.7 | 58.8 | 51.5 | 45.0 | 15.4 | 27.6 | 41.7 |
| | M.→D. | 69.1 | 63.4 | 79.8 | 48.5 | 67.2 | 44.3 | 62.1 | 67.2 | 63.3 | 80.2 | 53.6 | 64.7 | 49.5 | 63.4 |
| | M.→D.→W. | 73.9 | 65.5 | 78.4 | 57.9 | 69.1 | 53.4 | 66.4 | 71.4 | 63.5 | 79.9 | 57.9 | 70.1 | 49.8 | 65.4 |
| GUI-Actor (Wu et al., 2026) | Upper Bound | - | - | - | - | - | - | - | 96.9 | 89.6 | 97.4 | 86.4 | 95.7 | 84.7 | 92.5 |
| | M. | 74.4 | 76.4 | 58.8 | 61.4 | 32.2 | 63.1 | 61.0 | 76.7 | 77.5 | 61.1 | 61.9 | 35.0 | 67.1 | 63.2 |
| | M.→D. | 91.2 | 71.2 | 85.1 | 70.0 | 74.4 | 69.4 | 76.9 | 95.4 | 73.3 | 85.0 | 73.8 | 75.4 | 68.8 | 78.6 |
| | M.→D.→W. | 87.6 | 70.7 | 87.1 | 67.1 | 72.6 | 65.1 | 75.0 | 91.1 | 74.9 | 87.8 | 72.2 | 80.0 | 61.2 | 77.9 |
| *RFT-based GUI Method* | | | | | | | | | | | | | | | |
| SE-GUI† (Yuan et al., 2026; Liu et al., 2025b) | Upper Bound | - | - | - | - | - | - | 88.2 | - | - | - | - | - | - | 90.3 |
| | M. | 91.5 | 75.9 | 72.9 | 59.3 | 73.9 | 55.1 | 71.4 | 90.4 | 81.0 | 84.1 | 60.1 | 71.8 | 64.5 | 75.3 |
| | M.→D. | 91.8 | 76.6 | 78.1 | 65.0 | 76.3 | 58.3 | 74.4 | 92.2 | 81.9 | 87.6 | 67.4 | 73.7 | 63.5 | 77.7 |
| | M.→D.→W. | 92.0 | 78.2 | 83.1 | 65.7 | 77.4 | 60.2 | 76.1 | 93.2 | 82.5 | **90.2** | 69.3 | 75.1 | 63.1 | 78.9 |
| GUI-AiF | M. | 91.4 | 77.3 | 83.5 | 67.1 | 76.9 | 63.5 | 76.6 | 91.3 | 81.1 | 88.1 | 70.7 | 76.1 | 67.9 | 79.2 |
| | M.→D. | 92.7 | 78.2 | 86.6 | 68.4 | 76.2 | 62.2 | 77.4 | 94.1 | 81.4 | 88.7 | 72.1 | 76.7 | 66.1 | 79.9 |
| | M.→D.→W. | 91.9 | **79.5** | 87.1 | **70.7** | 78.2 | 65.8 | 78.9 | 94.8 | 82.0 | **90.2** | 75.0 | 76.8 | 68.2 | 81.2 |
| GUI-G² (Tang et al., 2026) | Upper Bound | 96.7 | 90.8 | 95.9 | 88.6 | 90.9 | 86.9 | 92.0 | - | - | - | - | - | - | 93.3 |
| | M. | 91.9 | 77.2 | 78.8 | 61.4 | 72.6 | 51.2 | 72.2 | 92.5 | 81.5 | 85.2 | 66.4 | 73.5 | 64.8 | 77.3 |
| | M.→D. | 93.8 | 76.4 | 88.2 | 61.4 | 75.2 | 53.9 | 74.8 | 94.2 | 78.7 | 87.2 | 66.4 | 77.8 | 65.7 | 78.3 |
| | M.→D.→W. | 95.2 | 76.4 | 92.8 | 63.6 | 79.6 | 54.9 | 77.1 | 94.8 | 79.1 | 89.3 | 68.6 | 81.2 | 68.6 | 80.3 |
| GUI-AiF | M. | 92.9 | 78.7 | 81.9 | 68.6 | 81.6 | 65.1 | 78.1 | 93.3 | 82.1 | 88.5 | 73.6 | 80.3 | 68.0 | 80.9 |
| | M.→D. | 94.1 | 77.3 | 93.2 | 69.3 | 81.9 | 66.2 | 80.3 | 95.9 | 79.6 | 89.7 | 77.1 | 81.2 | 68.3 | 81.9 |
| | M.→D.→W. | **96.1** | 76.9 | **95.7** | 68.3 | **84.8** | **68.2** | **81.7** | **96.6** | **83.6** | **90.2** | **77.4** | **82.9** | **70.5** | **83.5** |

# 4. Experiments

## 4.1. Continual GUI Agents Implementation

Unlike typical GUI agent training pipelines that assume static environments, we propose two dynamic GUI scenarios for continual learning (domain and resolution sequences), as shown in Table 1.

For domain sequence, we start to train GUI agents with Widget Captioning (Cheng et al., 2024) for mobile applications, and then separately train on the desktop and web data from ShowUI-web (Lin et al., 2025). For resolution sequence, we first train on ShowUI-web (Lin et al., 2025) (containing 1080p resolution desktop/web data), then we sequentially train on OmniACT (Kapoor et al., 2024) which includes desktop/web data resolution from 1080p up to 4K.

We evaluate on three benchmarks: ScreenSpot-V1 (Cheng et al., 2024) (SSv1), ScreenSpot-V2 (Wu et al., 2025) (SSv2) and ScreenSpot-Pro (Li et al., 2025a) (SSPro). The former two cover mobile, desktop, and web tasks for continual domain evaluation, and the latter one contains 6 high-resolution software interfaces including CAD, Development Programming (Dev), Creative Software (Creative), Scientific and Analytic (Scientific), Office Software (Office) and Operating System Commons (OS) to evaluate continual resolution performance.

## 4.2. Baselines and Training Details

We consider the SeeClick (Cheng et al., 2024) and GUI-Actor (Wu et al., 2026) as SFT-based GUI agent methods. Among RFT-based SOTA methods approaches: InfiGUI-R1 (Liu et al., 2025b), SE-GUI (Yuan et al., 2026) and GUI-G² (Tang et al., 2026) employing distinct reward policies. InfiGUI-R1 only computes IoU rewards between predicted and ground truth bounding boxes, SE-GUI solely uses distance rewards between predicted and ground truth center points. GUI-G² models bounding boxes with Gaussian distributions to calculate dense rewards considering both center point and region coverage. Since GUI-AiF focuses on the reward of interaction points and element regions, we combine the IoU rewards of InfiGUI-R1 with distance rewards of SE-GUI into an integrated baseline named by SE-GUI†.

We leverage Qwen2.5VL-3B for fine-tuning (Bai et al., 2025) in all experiments. Besides, we refer to the best results from baselines as upper bound. Since they leverage more abundant training datasets, upper bounds usually obtain better performance. SeeClick (Cheng et al., 2024) use a

*Table 3.* Performance (%) in continual GUI resolution on ScreenSpot-Pro. The N. and H. represent the Normal and High resolution. The upper bound comes from the optimal results of traditional GUI Agent method and serves as a reference.

| Method | | SSPro Accuracy (%) | | | | | | | | | | | | Avg. |
|---|---|---|---|---|---|---|---|---|---|---|---|---|---|---|
| | | CAD | | Dev | | Creative | | Scientific | | Office | | OS | | |
| | | Text | Icon | Text | Icon | Text | Icon | Text | Icon | Text | Icon | Text | Icon | |
| *Proprietary Model* | | | | | | | | | | | | | | |
| GPT-4o (OpenAI, 2024) | | 2.0 | 0.0 | 1.3 | 0.0 | 1.0 | 0.0 | 2.1 | 0.0 | 1.1 | 0.0 | 0.0 | 0.0 | 0.8 |
| Claude Computer Use (Anthropic, 2024) | | 14.5 | 3.7 | 22.0 | 3.9 | 25.9 | 3.4 | 33.9 | 15.8 | 30.1 | 16.3 | 11.0 | 4.5 | 17.1 |
| *Open-source Model* | | | | | | | | | | | | | | |
| Qwen2.5VL-3B (Bai et al., 2025) | | 9.1 | 7.3 | 22.1 | 1.4 | 26.8 | 2.1 | 38.2 | 7.3 | 33.9 | 15.1 | 10.3 | 1.1 | 16.1 |
| *SFT-based GUI Methods* | | | | | | | | | | | | | | |
| | Upper Bound | 2.5 | 0.0 | 0.6 | 0.0 | 1.0 | 0.0 | 3.5 | 0.0 | 1.1 | 0.0 | 2.8 | 0.0 | 1.1 |
| SeeClick (Cheng et al., 2024) | N. | 23.4 | 3.1 | 16.2 | 0.7 | 12.1 | 2.1 | 18.1 | 11.8 | 29.4 | 7.5 | 5.6 | 3.4 | 11.1 |
| | N.→H. | 26.4 | 6.3 | 25.3 | 0 | 20.8 | 5.6 | 28.1 | 14.6 | 35.6 | 11.3 | 17.7 | 8.9 | 16.7 |
| *RFT-based GUI Methods* | | | | | | | | | | | | | | |
| | Upper Bound | 55.8 | 12.5 | 68.8 | 17.2 | 57.1 | 15.4 | 77.1 | 24.5 | 74.0 | 32.7 | 57.9 | 21.3 | 47.5 |
| GUI-G$^2$ (Tang et al., 2026) | N. | 24.3 | 1.7 | 17.5 | 2.1 | 20.2 | 6.3 | 22.2 | 12.8 | 36.2 | 11.3 | 17.7 | 4.5 | 14.7 |
| | N.→H. | 24.5 | 6.3 | 24.4 | **4.1** | **23.7** | 5.6 | 27.3 | 12.6 | 35.1 | 9.4 | 21.2 | 5.6 | 16.7 |
| GUI-AiF | N. | **27.4** | 4.7 | 24.7 | 1.4 | 17.7 | **8.2** | 29.2 | 14.6 | 35.3 | **20.8** | 17.8 | **10.1** | 17.7 |
| | N.→H. | 20.3 | **15.6** | **29.2** | 2.1 | 23.2 | 3.5 | **33.3** | **14.7** | **38.4** | 18.2 | **22.4** | 6.8 | **19.0** |

older Qwen2-VL model hence getting a worse upper bound. For training, we implement 4 NVIDIA A100-80G GPUs for one epoch with the following hyperparameters: learning rate 1e-6, global batch size 8 and 4 generated predictions for each instruction. We set KL divergence $\beta$ to 0.04, employ Flash Attention 2 and bfloat16 precision with gradient checkpointing. The weights $\alpha$ and $\gamma$ are set to 15 and 0.5.

### 4.3. Main Results

**Continual GUI Domain**. We first evaluate the continual domain performance of GUI-AiF on SSv1 and SSv2 shown in Table 2. We first find the proprietary model struggles with GUI tasks. In contrast, while open-source models present some capability in this area, their performance indicates space for further improvement. Secondly, SFT methods and RFT methods both can be leveraged into Continual GUI Agents, while the performance of SFT methods is generally poor, verifying its over-adaptation tendency in a single GUI task. While the RFT baselines show some adaptation, GUI-AiF further boosts continual performance based on these methods, demonstrating that GUI-AiF enhances the GUI agents' adaptability facing domain shifts, such as migrating from mobile OS to Web OS. Finally, the performance of single-domain generally increases after training on each task, indicating effective continual learning of GUI-AiF which accumulates knowledge from learned tasks.

**Continual GUI Resolution**. We then evaluate the continual resolution performance of GUI-AiF in Table 3. Since the Gaussian modeling-based RFT provides better performance, we select the GUI-G$^2$ (Tang et al., 2026) as the representative RFT baseline in this and following evaluations. Firstly, while proprietary Claude (Anthropic, 2024) and open-source

Qwen2.5VL-3B (Bai et al., 2025) show comparable general performance, they were not evaluated on the shifted resolution tasks. Crucially, when facing the normal to high resolution shift, both the SFT and the RFT baselines achieve the exact same final average accuracy and are worse than GUI-AiF. This indicates that neither above paradigms can inherently handle resolution variation. The finding implies a shift in resolution is a critical visual distortion for element location. By promoting generalization in these two aspects, our GUI-AiF enhances the GUI agents' performance on this challenge.

**GUI-AiF Rewards Ablation**. We conduct the ablation study of APR-iF (🖱) and ARR-iF (⤓) on SSv1, SSv2 and SSPro, as shown in Table 4 and Table 5. Firstly, the full GUI-AiF rewards achieve the best performance, outperforming single APR-iF (🖱) or ARR-iF (⤓). This indicates that rewarding perception ability of generalizing interaction locations and element scales is both core to Continual GUI Agents. We can observe the performance gain of full GUI-AiF rewards is decreased when either reward is removed.

Besides, we find the final Avg. accuracy of APR-iF (🖱) outperforms ARR-iF (⤓) under all three benchmarks, which demonstrates the APR-iF (🖱) reward has a more significant positive impact on performance. This impact a nature of the variation in GUI environment: task shifts such as migrating from Mobile to Web or scaling to a higher resolution, are a fundamental form of layout transformation that causes element positions to change drastically. The fact that APR-iF (🖱) which generalizes points, is better equipped to handle this variation. This experiment suggests that learning to adapt to new interaction locations is a more critical factor than adapting to element scales for Continual GUI Agents.

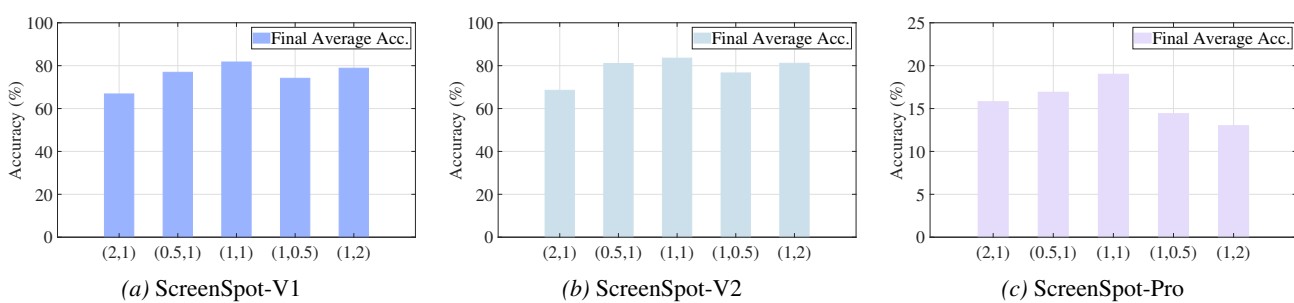

*Figure 3.* Hyperparameter sensitivity analysis for $\alpha$ and $\gamma$. Performance peaks at $(\alpha, \gamma) = (1,1)$ on three benchmarks.

*Table 4.* Ablation study of APR-iF (🖱) and ARR-iF (↳) on ScreenSpot-V1 and ScreenSpot-V2. The M., D., and W. represent the Mobile, Desktop, and Web platforms. The second and third lines refer to using APR-iF and ARR-iF, respectively.

| Method | | SSv1 Accuracy (%) | | | | | | | SSv2 Accuracy (%) | | | | | | |
| | | Mobile | | Desktop | | Web | | Avg. | Mobile | | Desktop | | Web | | Avg. |
| | | Text | Icon | Text | Icon | Text | Icon | | Text | Icon | Text | Icon | Text | Icon | |
| GUI-AiF | M. | 92.9 | **78.7** | 81.9 | 68.6 | 81.6 | 65.1 | 78.1 | 93.3 | 82.1 | 88.5 | 73.6 | 80.3 | 68.0 | 80.9 |
| | M.→D. | 94.1 | 77.3 | 93.2 | **69.3** | 81.9 | 66.2 | 80.3 | 95.9 | 79.6 | 89.7 | 77.1 | 81.2 | 68.3 | 81.9 |
| | M.→D.→W. | **96.1** | 76.9 | **95.7** | 68.3 | **84.8** | **68.2** | **81.7** | **96.6** | **83.6** | 90.2 | **77.4** | **82.9** | **70.5** | **83.5** |
| APR-iF (🖱) | M. | 91.6 | 77.8 | 81.9 | 67.1 | 80.1 | 63.1 | 76.9 | 93.1 | 82.9 | 86.0 | 71.4 | 78.6 | 67.5 | 80.0 |
| | M.→D. | 93.4 | 74.7 | 90.2 | 66.4 | 76.5 | 51.5 | 75.5 | 95.2 | 79.1 | 88.8 | 68.6 | 77.4 | 56.7 | 77.6 |
| | M.→D.→W. | 93.4 | 75.9 | 93.2 | 65.0 | 78.7 | 54.9 | 76.9 | 94.8 | 80.6 | 90.5 | 70.0 | 79.4 | 66.5 | 80.3 |
| ARR-iF (↳) | M. | 89.7 | 74.2 | 81.5 | 65.0 | 76.5 | 61.7 | 75.5 | 91.0 | 77.8 | 85.1 | 70.0 | 77.8 | 64.5 | 77.7 |
| | M.→D. | 92.7 | 75.5 | 89.2 | 62.9 | 73.9 | 49.0 | 75.4 | 94.1 | 78.2 | 88.1 | 67.9 | 75.2 | 54.2 | 76.3 |
| | M.→D.→W. | 92.3 | 76.9 | 90.7 | 63.6 | 75.2 | 50.5 | 76.4 | 94.1 | 79.1 | **90.7** | 67.9 | 78.2 | 56.7 | 77.8 |

*Table 5.* Ablation study of APR-iF (🖱) and ARR-iF (↳) on ScreenSpot-Pro. The N. and H. represent the Normal and High resolution platforms. The second and third lines refer to using APR-iF and ARR-iF, respectively.

| Method | Setting | CAD | | Dev | | Creative | | Scientific | | Office | | OS | | Avg. |
| | | Text | Icon | Text | Icon | Text | Icon | Text | Icon | Text | Icon | Text | Icon | |
| GUI-AiF | N. | 27.4 | 4.7 | 24.7 | 1.4 | 17.7 | **8.2** | 29.2 | 14.6 | 35.3 | **20.8** | 17.8 | **10.1** | 17.7 |
| | N.→H. | 20.3 | **15.6** | 29.2 | **2.1** | 23.2 | 3.5 | **33.3** | 14.7 | **38.4** | 18.2 | 22.4 | 6.8 | **19.0** |
| APR-iF (🖱) | N. | 30.5 | 7.8 | 24.7 | **2.1** | 24.8 | 2.1 | 25.2 | 10.0 | 31.8 | 15.1 | 23.4 | 4.5 | 16.8 |
| | N.→H. | **32.5** | 4.7 | 22.5 | 1.4 | **26.8** | 5.6 | 26.1 | 14.6 | 37.5 | 16.9 | **24.1** | 5.6 | 18.2 |
| ARR-iF (↳) | N. | 29.4 | 4.7 | 22.7 | 1.4 | 21.8 | 2.9 | 25.2 | 12.7 | 30.1 | 15.1 | 21.2 | 5.6 | 16.1 |
| | N.→H. | 28.9 | 6.3 | 31.2 | 0 | 23.3 | 3.5 | 28.9 | **16.4** | 36.3 | 13.2 | 23.6 | 4.5 | 18.0 |

### 4.4. Discussion

**Hyperparameter Sensitivity.** GUI-AiF leverages the weights $\alpha$ and $\gamma$ to control the intensity of exploration over diverse points and regions. We explore the effect of these two hyperparameters. Specifically, we scale two parameters to 2 times and 1/2, and present final average accuracy of (Tang et al., 2026) GUI-AiF on SSv1, SSv2, and SSPro in Figure 3. Firstly, we find that $\alpha$ which is related to APR-iF (🖱) has a greater impact on performance in the continual GUI domain scenario. We consider that domain flux is more reflected in its impact on GUI interaction points. In contrast, varying $\gamma$ expresses a more obvious impact under contin-

ual GUI resolution, potentially caused by higher resolution leads to the diversity of GUI element scales.

**Forward Transfer Evaluation.** We further find that GUI-AiF enhances forward transfer capability of GUI agents, enabling them to leverage knowledge learned from prior tasks to generalize to subsequent tasks. As the results shown in Figure 4, after training on Mobile platforms, the performance achieved on the subsequent Desktop and Web platforms by GUI-AiF surpasses the corresponding baseline under SSv2. Similar improvement can be observed under SSPro: the results trained on normal resolution task with GUI-AiF generally outperform corresponding baseline

*Table 6.* Ablation study w/o KL divergence under continual GUI domain and resolution. **Bold** represents the best result in each task.

| | SSv1 Accuracy (%) | | | | | | | SSv2 Accuracy (%) | | | | | | |
| | Mobile | | Desktop | | Web | | Avg. | Mobile | | Desktop | | Web | | Avg. |
| | Text | Icon | Text | Icon | Text | Icon | | Text | Icon | Text | Icon | Text | Icon | |
| M. | 83.5 | 70.3 | 79.4 | 66.4 | 71.3 | **58.7** | 71.6 | 83.8 | **83.9** | 82.5 | 70.0 | 67.5 | **65.5** | 73.9 |
| M.→D. | 91.9 | **75.5** | 86.1 | 64.3 | 72.6 | 48.5 | 73.2 | 92.6 | 76.3 | 88.1 | 67.9 | 74.9 | 55.2 | 75.8 |
| M.→D.→W. | **93.0** | 74.7 | **92.3** | **68.6** | **77.4** | 56.3 | **77.1** | 92.8 | 77.7 | **93.3** | **73.6** | **78.2** | 61.1 | **79.6** |

| | SSPro Accuracy (%) | | | | | | | | | | | | | |
| | CAD | | Dev | | Creative | | – | Scientific | | Office | | OS | | Avg. |
| | Text | Icon | Text | Icon | Text | Icon | | Text | Icon | Text | Icon | Text | Icon | |
| N. | 22.3 | 1.6 | 23.4 | **2.1** | 18.2 | 3.5 | – | 22.2 | 11.8 | 28.8 | **11.3** | 15.9 | 4.5 | 13.8 |
| N.→H. | **30.5** | **3.1** | **34.4** | 0.7 | **24.2** | **5.6** | – | **30.2** | **13.6** | **34.1** | 9.4 | **28.0** | **6.7** | **18.4** |

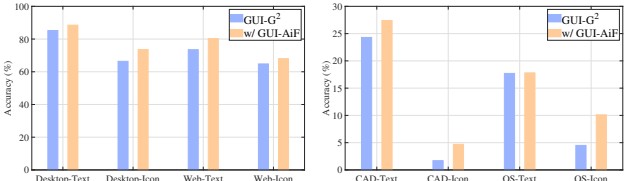

*(a)* Trained on mobile platform.  *(b)* Trained on normal resolution.

*Figure 4.* Forward transfer evaluation of GUI-AiF.

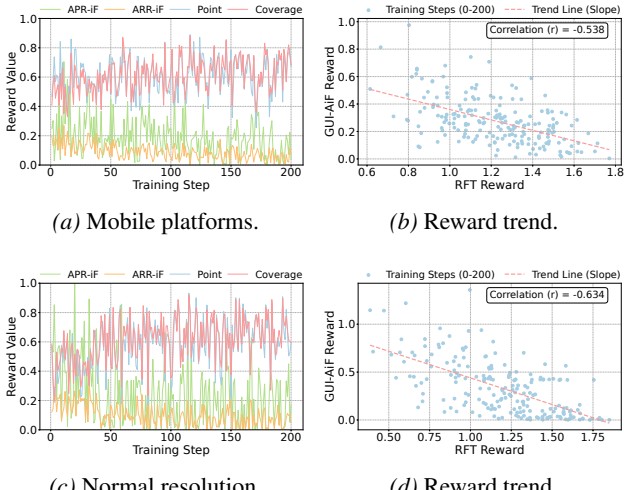

*(a)* Mobile platforms.  *(b)* Reward trend.

*(c)* Normal resolution.  *(d)* Reward trend.

*Figure 5.* Reward value analysis within two scenarios. (a)/(c) show reward statistics, while (b)/(d) show their trend distributions.

in high resolution task. These results show that encouraging GUI agents to explore diverse interaction regions prevents them from over-adapting to specific layouts, leading to stronger generalization when facing interface shifts. This observation provides favorable support for GUI-AiF.

**KL Divergence.** As the KL term is beneficial for providing continual capacity via measuring the gap from a reference model, we remove KL term in the ablation study and conduct evaluations on three benchmarks shown in Table 6. As hypothesized, removing degrades continual learning performance of GUI agents compared to Table 2, the final average and specific task accuracy generally appear decline. This result demonstrates that solely pursuing diverse solutions is insufficient, as the unconstrained exploration can collapse the policy and overwrite prior knowledge. The KL term provides a stable foundation, ensuring GUI-AiF is grounded and hence helping agents adapt to dynamic GUI tasks.

**Reward Comparison Analysis.** We analyze the reward values within the initial 200 training steps of mobile platforms and normal resolution tasks. These rewards include APR-iF (🖱), ARR-iF (⬇), the RFT policy's rewards for predicted bounding box point and region coverage. From Figures 5a and 5c, we can observe that APR-iF (🖱) fluctuates more significantly than ARR-iF (⬇) within each task. We consider that exploration of interaction points plays a more critical role in the Continual GUI Agents, a finding that aligns with Table 4. Besides, we count the rewards value (APR-iF 🖱 + ARR-iF ⬇) of GUI-AiF and rewards

value (point + coverage) of RFT in each training step, and illustrate their changing trend by a scatter plot. As shown in Figures 5b and 5d, this indicate an inverse relationship between the two reward types: rewards designed to promote the generalization of GUI agents and those focused on optimizing performance on static current task, which appear to be conflicting objectives.

**Comparison with Common Continual Learning Methods.** We additional compare GUI-AiF with two common continual learning methods (data reply and parameter regularization), and evaluate them on SSv1 and SSv2 as the representative. Firstly, we conduct data reply by adding part of data from previous tasks into new task (Rebuffi et al., 2017). We then choose Lwf (Li & Hoiem, 2017) as parameter regularization, and build a baseline for SFT+LwF by measuring differences between training model and base model using KL terms (RFT naturally has KL items, so this method only can be integrated into SFT). Results are shown in Table 7, that indicate that common methods are unable to

*Table 7.* Comparison with common continual techniques in M.→D.→W. scenario. **Bold** represents the best result.

| Methods | SSv1 Accuracy (%) | | | | | | | SSv2 Accuracy (%) | | | | | | |
|---|---|---|---|---|---|---|---|---|---|---|---|---|---|---|
| | Mobile | | Desktop | | Web | | Avg. | Mobile | | Desktop | | Web | | Avg. |
| | Text | Icon | Text | Icon | Text | Icon | | Text | Icon | Text | Icon | Text | Icon | |
| Reply (Rebuffi et al., 2017) | 93.4 | 72.1 | 87.1 | 61.4 | 81.3 | 58.3 | 75.6 | 94.8 | 76.3 | 86.6 | 67.9 | 82.5 | 61.6 | 78.3 |
| Lwf (Li & Hoiem, 2017) | 92.3 | 75.1 | 86.1 | 55.7 | 77.8 | 55.3 | 73.7 | 91.7 | 78.2 | 84.0 | 60.7 | 76.5 | 55.2 | 74.4 |
| GUI-AiF | **96.1** | **76.9** | **95.7** | **68.3** | **84.8** | **68.2** | **81.7** | **96.6** | **83.6** | **90.2** | **77.4** | **82.9** | **70.5** | **83.5** |

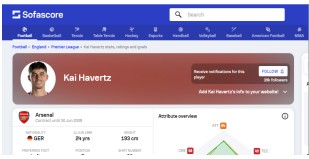

*(a)* Original Web interface.

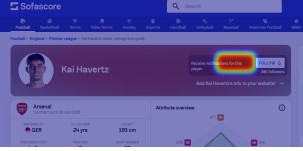

*(b)* SFT learns all domains.

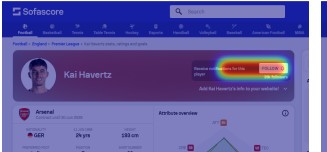

*(c)* GUI-AiF trains on Mobile.

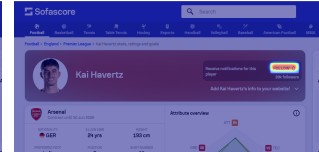

*(d)* GUI-AiF learns all domains.

*Figure 6.* Visualizations of interaction region heatmaps on web platforms. Instruction used: "Follow this player."

achieve effective continual performance for GUI agents.

**Visualization Comparison.** We provide the interaction region heatmaps shown in Figure 6. Figure 6a displays the Web interface with the instruction "Follow this player". First, the heatmap of SFT baseline learning all domain tasks (Figure 6b) fails to locate the correct interaction region, indicating the method's poor ability to adapt to the variation of shifting GUI interface. We then compare the heatmaps from GUI-AiF trained only on the Mobile domain (Figure 6c) versus GUI-AiF that has continually learned all domains (Figure 6d). We observe that while the former correctly identifies the interaction region, its bounding box remains unrelated to the content and lacks fine-grained precision, the latter accurately locates the region based on the instruction. This evolution demonstrates GUI-AiF's capability to refine its grounding through continual learning in shifting tasks.

**Limitations.** Due to the computing resource constraints, we mainly explore a 3B model scale, three specific training datasets, and three evaluation benchmarks; this scope inherently restricts the absolute performance of the GUI agents. We believe that expanding the model size and using more abundant data would enable the agents to further bridge the gap with the grounding boundary. Additionally, our exploration is in two types of GUI shifts (domain and resolution). However, the flux of real-world digital applications is far more diverse, such as layout shifts (app updates),

appearance shifts (dark mode), and localization (language changes). Our future work continues to explore the Continual GUI Agents under broader and more complex scenarios.

## 5. Conclusion

In this work, we formalize Continual GUI Agents, a new task setting that models GUI interaction under domain and resolution shifts, and we highlight the limitations of static grounding reward optimization in dynamic GUI environments. Then, we propose GUI-AiF, equipped with APR-iF (🖱) and ARR-iF (↪), which allow agents to continuously anchor interaction points and regions despite evolving GUIs, thereby mitigating over-adaptation to static grounding cues. Extensive experiments show that GUI-AiF outperforms state-of-the-art baselines, establishing the first continual learning framework for GUI agents.

## Impact Statement

This paper presents work whose goal is to advance the field of Machine Learning. There are many potential societal consequences of our work, none which we feel must be specifically highlighted here.

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

# Supplementary Material: Continual GUI Agents

## Abstract

*We provide four additional results in supplementary material:*
*(I) Training all the GUI datasets we leverage at once, investigating the upper boundary of GUI-AiF.*
*(II) Exploring the reversing sequence of continual domain tasks.*
*(III) Bias finding between Text and Icon Interaction.*
*(IV) Providing more GUI visualization heatmaps.*

## A. Upper Boundary of GUI-AiF

Since the boundaries provided in main paper are choosen from the best results from references, which uses larger model size and more extensive datasets. We analyze our GUI-AiF boundary by training our used dataset at once, and conduct evaluations in SSv1 and SSv2. The results is shown in Table 8 (represented by GUI-AiF$^*$). It shows the upper boundary is comparable to the performance achieved by sequentially continual learning all domain tasks, it verifies the effectiveness of GUI-AiF in continual GUI agents.

## B. Reversed Continual Domain Tasks

We then train domain tasks with the sequence of Web, Desktop and Mobile shown in Table 8 (represented by GUI-AiF$^\dagger$). We can find the performance of continual learning tasks still aligns with the tendency in the main paper.

## C. Bias between Text and Icon Interaction.

Beyond evaluations, our results generally demonstrate a GUI agents' bias towards text related interaction. From the Figure 7, it is evident that performance on text-based tasks is consistently higher than icon-based tasks, regardless of the domain or resolution. We consider this is due to a fundamental bias in the agents' visual branch, which defaults to its strong OCR capability for text comprehension, whereas it struggles with icons which are semantically ambiguous and lack this direct textual grounding.

## D. Visualizations of Continual GUI Agents

We additionally provide some interaction region heatmaps of continual domains and resolutions, they demonstrate the changes in GUI agents' perception of interaction interfaces based on instructions during the continual learning process.

*Table 8.* Performance (%) of upper boundary (GUI-AiF$^*$) and reversed tasks (GUI-AiF$^\dagger$) on ScreenSpot-V1 and ScreenSpot-V2. The M., D., and W. represent the Mobile, Desktop, and Web platforms. Gray are results in the main paper for reference. **Bold** represents the best result.

| Method | | SSv1 Accuracy (%) | | | | | | | SSv2 Accuracy (%) | | | | | | |
| | | Mobile | | Desktop | | Web | | Avg. | Mobile | | Desktop | | Web | | Avg. |
| | | Text | Icon | Text | Icon | Text | Icon | | Text | Icon | Text | Icon | Text | Icon | |
|---|---|---|---|---|---|---|---|---|---|---|---|---|---|---|---|
| GUI-AiF$^*$ | Upper Boundary | 92.3 | **80.1** | 94.7 | 68.6 | 83.3 | 67.4 | 81.1 | **97.1** | 83.2 | 92.3 | **79.3** | 84.9 | 66.2 | **83.8** |
| GUI-AiF | M. | 92.9 | 78.7 | 81.9 | 68.6 | 81.6 | 65.1 | 78.1 | 93.3 | 82.1 | 88.5 | 73.6 | 80.3 | 68.0 | 80.9 |
| | M.→D. | 94.1 | 77.3 | 93.2 | **69.3** | 81.9 | 66.2 | 80.3 | 95.9 | 79.6 | 89.7 | 77.1 | 81.2 | 68.3 | 81.9 |
| | M.→D.→W. | **96.1** | 76.9 | **95.7** | 68.3 | **84.8** | 68.2 | **81.7** | 96.6 | **83.6** | 90.2 | 77.4 | 82.9 | 70.5 | 83.5 |
| GUI-AiF$^\dagger$ | W. | 94.9 | 74.7 | 82.0 | 67.9 | 81.3 | 64.1 | 77.5 | 95.2 | 77.3 | 82.5 | 72.7 | 81.2 | 64.0 | 78.8 |
| | W.→D. | 95.6 | 69.4 | 87.6 | 67.1 | 83.7 | 64.8 | 78.1 | 95.9 | 72.5 | 88.7 | 76.7 | 77.8 | 69.6 | 80.2 |
| | W.→D.→M. | 89.7 | 72.6 | 92.6 | 69.6 | 81.4 | **69.5** | 79.2 | 89.7 | 74.9 | **92.7** | 72.7 | **86.8** | **72.2** | 81.5 |

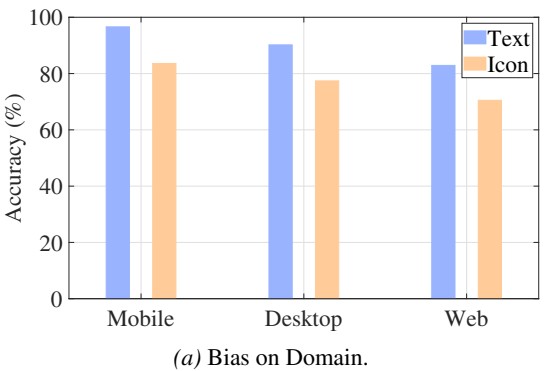
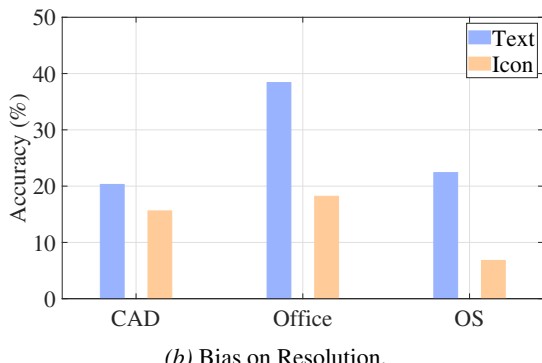

*(a)* Bias on Domain.

*(b)* Bias on Resolution.

*Figure 7.* GUI performance bias between text and icon.

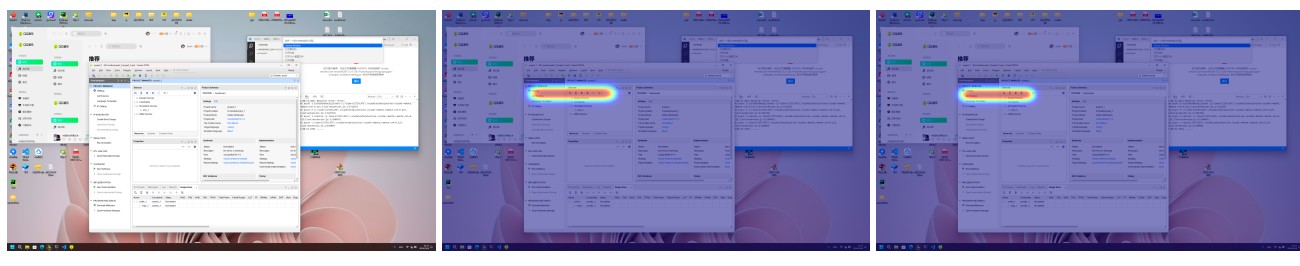

*(a)* Original Vivado interface.  *(b)* GUI-AiF trains on normal resolution.  *(c)* GUI-AiF learns all resolutions.

*Figure 8.* Visualizations of interaction region heatmaps on high resolution Vivado interface. Instruction used: "Add sources in Vivado."

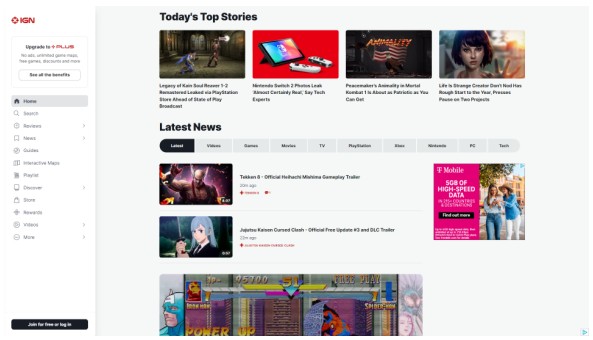
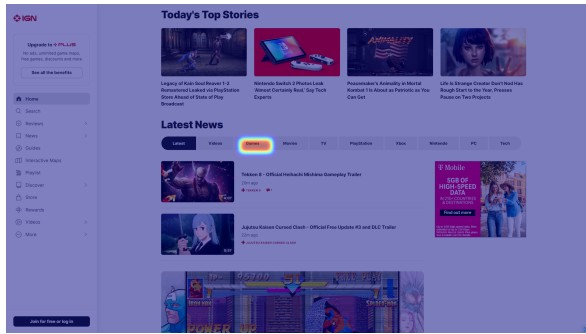

*(a)* Original Desktop interface.

*(b)* GUI-AiF trains on Mobile.

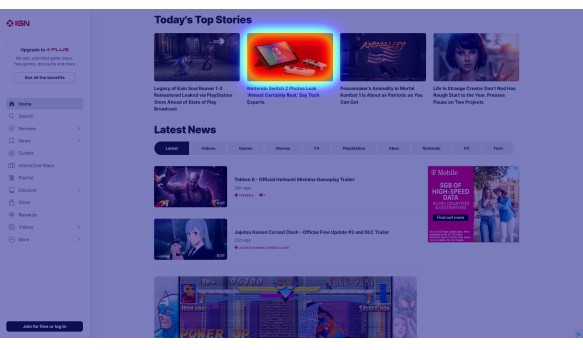
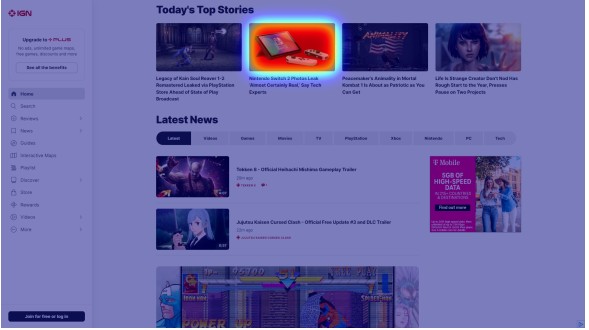

*(c)* GUI-AiF trains on Desktop.

*(d)* GUI-AiF learns all domains.

*Figure 9.* Visualizations of interaction region heatmaps on desktop platforms. Instruction used: "Find the Switch game console."

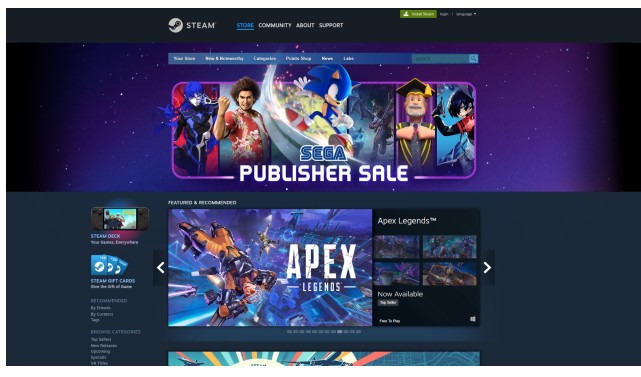

*(a)* Original Web interface.

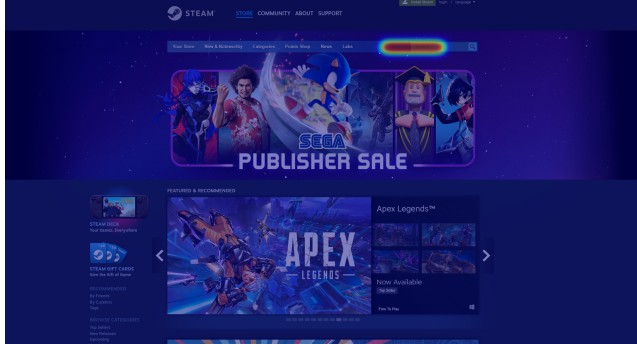

*(b)* GUI-AiF trains on Mobile.

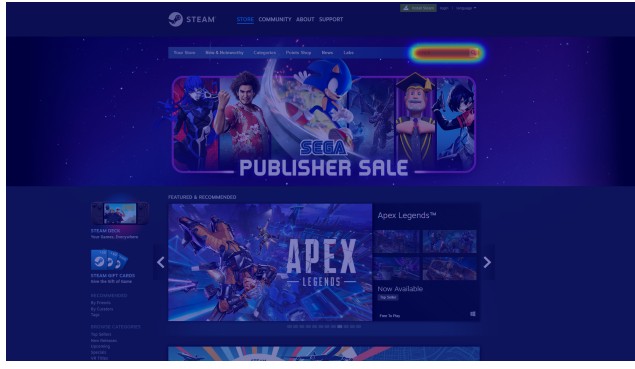

*(c)* GUI-AiF trains on Desktop.

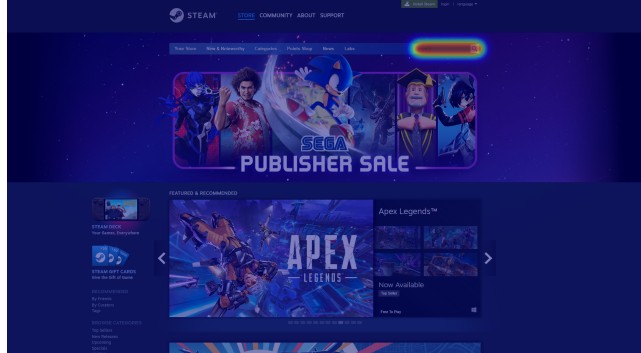

*(d)* GUI-AiF learns all domains.

*Figure 10.* Visualizations of interaction region heatmaps on web platforms. Instruction used: "Search a new game."

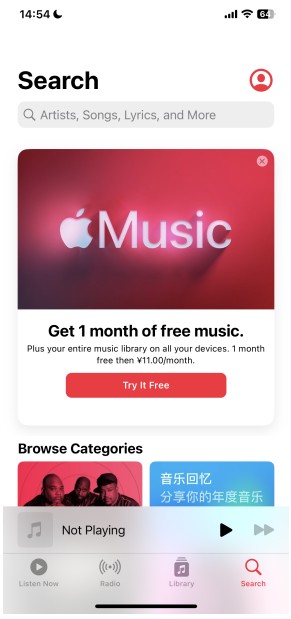

*(a)* Original Mobile interface.

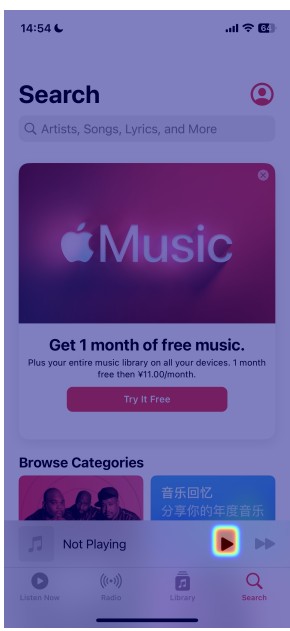

*(b)* GUI-AiF trains on Mobile.

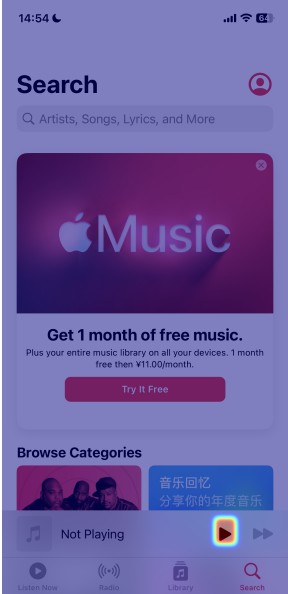

*(c)* GUI-AiF trains on Desktop.

*(d)* GUI-AiF learns all domains.

*Figure 11.* Visualizations of interaction region heatmaps on mobile platforms. Instruction used: "Play a song."

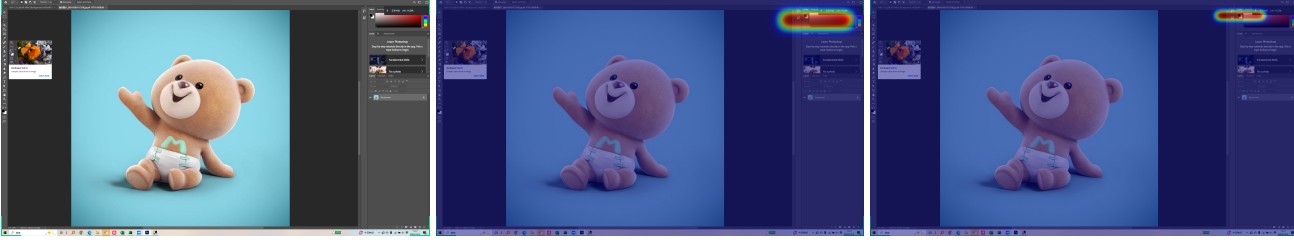

*(a)* Original Photoshop interface.     *(b)* GUI-AiF trains on normal resolution.     *(c)* GUI-AiF learns all resolutions.

*Figure 12.* Visualizations of interaction region heatmaps on high resolution Photoshop interface. Instruction used: "Select the color picker."

