# OpenReview forum: "Continual GUI Agents"
_ICML.cc/2026/Conference — ICML 2026 regular_

### Official Review · Reviewer_1g15 · 2026-03-11

**Soundness:** 3
**Presentation:** 4
**Significance:** 3
**Originality:** 3
**Overall Recommendation:** 4
**Confidence:** 4

**Summary:**

This paper proposes a novel task setting, Continual GUI Agents, to investigate the continuous learning problem of GUI agents under conditions of domain migration and resolution changes. The authors observe that existing SFT and RFT-based methods overfit to static grounding cues during environment drift, leading to performance degradation. To address this issue, the authors propose APR-iF, which is oriented towards interaction point diversity, and ARR-iF, which is oriented towards element region separation, within the GRPO pipeline. These methods alleviate overfitting to a single task by encouraging prediction diversity. Extensive experiments on three types of ScreenSpots effectively demonstrate that the proposed reward outperforms existing baseline methods in cross-domain and cross-resolution continuous learning scenarios.

**Compliance With Llm Reviewing Policy:**

Affirmed.

**Final Justification:**

The author has solved the problem I raised, and I am maintaining my positive score.

**Key Questions For Authors:**

* What does the GUI-AiF demonstration in Tables 2 and 3 mean? Does it refer to adding the proposed reward to the learner based on SE-GUI and GUI-G2 respectively, and then training? In this case, is r_i in Equation 5 the definition of the baseline RFT?

* Table 2 shows that the SFT method experiences a near-universal performance degradation from M.→D.→W. Is this a problem with SFT, or with the authors' continuous learning path? How can this be explained? Is there any discussion or ablation regarding the order of domain transfers and the relationships between domains?

* Regarding the domain order issue, the authors' experimental setup always involves transferring from simpler domains to more difficult domains. What would happen if it were reversed, for example, W.→D.→M., or H.→N.?

* How do the ablation experiments of the α and γ hyperparameters correspond to the final settings? The paper states that "The weights α and γ are set to 15 and 0.5," but from the ablation discussion, a larger proportion of α results in worse performance. Does (α, γ) = (a,b) in Figure 3 refer to a ratio or specific values? Based on the ablation trend, how was the final set value determined?

* Can the proposed reward function within an RL algorithm framework outside of GRPO?

**Limitations:**

* Lacks comparison with existing continuous learning algorithms. Since this is about continuous learning, it shouldn't be discussed solely with existing RFT methods.

* The experiments are plentiful, but some key ablation techniques and discussions are missing. See Weakness and Questions for details.

**Strengths And Weaknesses:**

### Strengths
* The problem definition is valuable, systematically proposing the continuous learning problem of GUI agents for the first time, defining two meaningful scenarios (cross-domain and cross-resolution), filling a research gap and providing guidance for subsequent research.
* The writing is logically coherent; the problem, motivation, and algorithmic innovations are easy to understand and mutually reinforcing.
* The experimental analysis is rich, not only conducting experiments and analyses on several existing SFT and RFT methods to address the problem, but also performing extensive ablation experiments, hyperparameter sensitivity analysis, forward transfer evaluation, reward comparison analysis, and visualization heatmaps, providing multi-faceted evidence for understanding the methods.

### Weaknesses
* The reward, in essence, encourages greater model divergence, unlike the design of existing RFT GUI agents. This divergence may promote the transferability mentioned by the authors, but will it not affect the accuracy of tasks within a single domain? How to balance diversity and accuracy? There is no clear explanation as to why "encouraging prediction diversity" can truly solve the catastrophic forgetting problem in continuous learning.
* This paper focuses on transfer learning, but its innovation lies mainly in the design of the RL reward. The experiments primarily compare other RFT models and algorithms, lacking comparisons with Continuous Learning methods such as Elastic Weight Consolidation and Experience Replay, thus lacking persuasiveness in the field of continuous learning.
* Judging from the ablation experiments in Table 4 and the discussion of hyperparameters, APR-iF seems to account for the main source of performance, while the importance and role of ARR-iF are not significant.
* Why was the ablation of the reward only performed on SSV1 and SSV2 (Table 4) and not on SSPro? The paper also mentions that ARR-iF is more important in scenarios with varying resolution; why wasn't this proven on SSPro?

---

> ### Author Rebuttal · Authors · 2026-03-31
>
> **Q1: How to balance diversity and accuracy. How to solve forgetting.**
> 1. Accuracy advantage and diversity reward are computed on these same $N$ outputs. Blindly maximizing either reward in isolation inevitably penalizes the other, the **correctness constraint for diversity is implicitly inherited through the accuracy reward.**
> 2. Refer to Reviewer jNMm's comments and [1, 2], the GUI-AiF reward is an **entropy maximization policy** specialized in GUI grounding task. This policy encourages exploration by **keeping the entropy over action high to mitigate forgetting.**
>
> [1] Function optimization using connectionist reinforcement learning algorithms. 1991.
> [2] Asynchronous methods for deep reinforcement learning. ICML 2016.
>
> **Q2 Q10: Lacking common CL methods**
> We leverage **data reply** and **parameter regularization** as common CL methods, evaluate them on SSv1 and SSv2 as the representative.
> Specific results refer to rebuttal in **Reviewer 4pST’s Q2 Q5.**
>
> **Q3: Importance and role of ARR-iF**
> ARR-iF’s importance is **more evident in benchmark with significant differences in element box scale.** To verify this, we select samples with significant differences in element box scale from mobile, desktop, and web subsets of SSv2. The results are as follows:
>
> |Method|Setting|Mobile| |Desktop| |Web| |Overall|
> |:---|:---|:---|:---|:---|:---|:---|:---|:---|
> | | |Text|Icon|Text|Icon|Text|Icon|Avg.|
> |GUI-AiF|M.→D.→W.|98.1|85.3|92.7|80.2|81.2|74.9|85.4|
> |APR-iF| M.→D.→W. |93.2|83.3|90.8|70.0|80.6|68.7|81.1|
> |ARR-iF| M.→D.→W. |95.7|80.1|91.3|73.5|79.7|64.5|80.8|
>
> **Q4 Q11: Reward ablation on SSPro**
> We provide the results as following:
>
> |Method|Setting|CAD| |Dev| |Creative| |Scientific| |Office| |OS| | |
> |:---|:---|:---|:---|:---|:---|:---|:---|:---|:---|:---|:---|:---|:---|:---|
> | | |Text|Icon|Text|Icon|Text|Icon|Text|Icon|Text|Icon|Text|Icon|Avg.|
> |GUI-AiF|N.|27.4|4.7|24.7|1.4|17.7|**8.2**|29.2|14.6|35.3|**20.8**|17.8|**10.1**|17.7|
> | |N.→H.|20.3|**15.6**|29.2|**2.1**|23.2|3.5|**33.3**|14.7|**38.4**|18.2|22.4|6.8|**19.0**|
> |APR-iF|N|30.5|7.8|24.7|**2.1**|24.8|2.1|25.2|10.0|31.8|15.1|23.4|4.5|16.8|
> | |N.→H.|**32.5**|4.7|22.5|1.4|**26.8**|5.6|26.1|14.6|37.5|16.9|**24.1**|5.6|18.2|
> |ARR-iF|N.|29.4|4.7|22.7|1.4|21.8|2.9|25.2|12.7|30.1|15.1|21.2|5.6|16.1|
> | |N.→H.|28.9|6.3|**31.2**|0|23.3|3.5|28.9|**16.4**|36.3|13.2|23.6|4.5|18.0|
>
> The results match with our analysis in manuscript.
>
> **Q5: GUI-AiF demonstration in Tables**
> The GUI-AiF in Tables represents the reward integrated into different baselines, and in Formula 5, $r_i$ is the reward value in the definition of the baseline RFT. Essentially, any RFT-based GUI agent that formulates interaction points and/or regions can use GUI-AiF reward to enhance its continual adaptation across diverse UI environments.
>
> **Q6 Q7: SFT performance and domain transfers path.**
> The method experiencing degradation is GUI-Actor, which is modified on the SFT training paradigm. **This result depends on the specific method used in the paper.** Given that SeeClick is a pure SFT method, its results demonstrate a consistent trend with our analysis. Regarding the domain transfers path, **we has provided a reversed continual learning path W.→D.→M. at Table 6 in Supplementary Material,** the results show that GUI-AiF has a stable performance.
>
> **Q8: Hyperparameters setting.**
> (a,b) in Figure 3 represents the **scaling ratios** applied to our default values of 15 and 0.5. our experiments demonstrate that setting these hyperparameters either too large or too small degrades overall performance. Therefore, the values 15 and 0.5 were finally determined.
>
> **Q9: GUI-AiF applied outside of GRPO?**
> Theoretically, yes. We can integrate GUI-AiF into PPO and generate multiple outputs, but doing so will incur significant computational complexity.

---

> > ### Author Rebuttal · Reviewer_1g15 · 2026-04-02
> >
> > The author has addressed my main concerns. I have no further questions.

---

> > > ### Author Response · Authors · 2026-04-04
> > >
> > > We sincerely appreciate your suggestions and reply.

---

### Official Review · Reviewer_62uh · 2026-03-11

**Soundness:** 2
**Presentation:** 3
**Significance:** 3
**Originality:** 2
**Overall Recommendation:** 4
**Confidence:** 3

**Summary:**

The paper introduces the task of "Continual GUI Agents," focusing on the performance degradation of GUI agents when encountering new domains (e.g., Mobile to Web) or resolution shifts (e.g., 1080p to 4K). To address this, the authors propose **GUI-AiF**, a Reinforcement Fine-Tuning (RFT) framework. It incorporates two novel reward mechanisms: Anchoring Point Reward in Flux (APR-iF) and Anchoring Region Reward in Flux (ARR-iF). APR-iF rewards the spatial variance of predicted interaction points to encourage exploration, while ARR-iF uses Bhattacharyya distance between Gaussian-modeled bounding boxes to promote diversity in predicted scales. The method is evaluated on ScreenSpot-V1, V2, and Pro benchmarks, demonstrating improvements over static supervised fine-tuning (SFT) and standard RFT baselines.

**Compliance With Llm Reviewing Policy:**

Affirmed.

**Final Justification:**

I have read the authors’ rebuttal and the subsequent discussion. The rebuttal adequately addressed my main concerns. Accordingly, I have raised my score by 1 point.

**Key Questions For Authors:**

1. Regarding the $N$ predictions used for advantage estimation: are these generated through diverse rollouts (e.g., high temperature) or deterministic sampling? If the latter, how is the variance in APR-iF meaningful?
2. How is the reference model $\pi_{ref}$ defined? Is it the initial pre-trained model or the model frozen after the *previous* task? This distinction is vital for determining if the KL-divergence effectively mitigates forgetting or simply acts as a local trust-region constraint.
3. Why was the Bhattacharyya distance chosen over other metrics like IoU-based diversity? Specifically, how does the model avoid the "degeneration" where it produces wildly inaccurate boxes just to maximize the $D_B$ reward?
4. Can the authors provide a comparison against a simple replay-based baseline or EWC to verify if GUI-AiF offers unique advantages over standard continual learning techniques?

If the above questions are resolved, my score will increase.

**Limitations:**

The authors acknowledge limitations regarding model scale (3B) and the breadth of GUI shifts. However, the discussion should be expanded to include the marginal performance gains in high-resolution scenarios and the potential for the model to generate "hallucinated" diverse regions to satisfy the ARR-iF reward without improving actual grounding accuracy.

**Strengths And Weaknesses:**

**Strengths:**
1. **Overall, the authors discuss a pressing issue:** The brittleness of GUI agents in the face of evolving digital environments and UI layouts is a critical bottleneck for real-world deployment.
2. The adoption of the Reinforcement Fine-Tuning (RFT) paradigm is well-motivated, as on-policy updates with KL-regularization are inherently more suitable for maintaining stability during distribution shifts than standard SFT.
3. **Overall, this study's significant contribution comprises** a specialized reward shaping strategy (APR-iF and ARR-iF) designed to mitigate over-adaptation to specific coordinates and scales, which is a common failure mode in GUI grounding.

**Weaknesses:**
1. **Inaccurate Novelty Claims:** The claim of being the "first continual learning framework for GUI agents" is factually incorrect. Several prior works have addressed continual or adaptive learning for GUI/Vision-Language tasks, such as UIShift [1], LLaVA-c [2], and ZeroFlow [3]. The authors fail to position their work relative to these existing "flux" or continual learning scenarios.
2. **Methodological Logic Gaps:** The link between APR-iF (rewarding high variance of predicted points) and the mitigation of catastrophic forgetting is not clearly established. High variance among predictions for the same instruction could signify stochastic noise or model instability rather than a "generalizable" grounding capability. Furthermore, $\mathcal{R}_p$ lacks an inherent correctness constraint; its interaction with the accuracy-based advantage $A_i$ requires a more rigorous formal analysis.
3. **Superficial Literature Review:** Section 2.2 lacks depth regarding the classic landscape of Continual Learning. The paper ignores fundamental strategies such as Elastic Weight Consolidation (EWC) [4], Learning without Forgetting (LwF) [5], or replay-based methods, which are standard baselines in this field.
4. **Mathematical Flaws in Gaussian Modeling:** Modeling axis-aligned bounding boxes (AABB) as 2D Gaussians for ARR-iF is problematic. The use of a diagonal covariance matrix $\Sigma_i$ assumes independence between width and height, which is rarely true for UI elements (e.g., specific aspect ratios for sidebars or buttons). Additionally, the authors confuse the *area* of a box with a probability distribution, which might lead to trivial solutions where the model simply increases variance to maximize distance rather than learning meaningful scale diversity.
5. **Hyperparameter Sensitivity:** The weights $\alpha=15$ and $\gamma=0.5$ show a 30-fold imbalance. The paper does not provide a normalization strategy or a robustness analysis of these scales across different datasets/resolutions.
6. **Marginal Performance Gains:** On the SSPro benchmark (N.→H.), the improvement over SeeClick is only 2.3% (16.7% vs 19.0%). Given that SeeClick was not designed for continual learning, this marginal gain raises questions about the actual effectiveness of the proposed specialized rewards.

[1] Gao et al., "UIShift: Enhancing VLM-based GUI Agents through Self-supervised Reinforcement Learning," 2025.

[2] Liu et al., "LLaVA-c: Continual Improved Visual Instruction Tuning," 2025.

[3] Feng et al., "ZeroFlow: Overcoming Catastrophic Forgetting is Easier than You Think," 2025.

[4] Kirkpatrick et al., "Overcoming catastrophic forgetting in neural networks," 2017.

[5] Li & Hoiem, "Learning without Forgetting," 2018.

---

> ### Author Rebuttal · Authors · 2026-03-31
>
> **Q1: Inaccurate Novelty Claims**
> LLaVA-c handles visual instruction tuning degradation (**e.g., VQA**), where the model overfits to single-task instructions (e.g., answering in a single word).
> ZeroFlow is on **classification-based continual learning**, it does not involve GUI tasks.
> UIShift addresses the data bottleneck in GUI agent training by employing an instruction-free approach. **It is still a training framework for static UI environments.**
> GUI-AiF focuses on preventing spatial overfitting of the model on specific UI data, enhancing GUI grounding’s adaption in shifted domain and resolution scenarios.
> Overall, the listed papers are out of our scope in terms of the tasks and scenarios.
>
> **Q2: Methodological Logic Gaps**
> The GUI-AiF (APR-iF and ARR-iF) reward is a **entropy maximization policy** specialized in GUI grounding task, this policy encourages exploration by **keeping the entropy over action high to mitigate forgetting (refer to Reviewer jNMm's comments and [1, 2]).**
> For reward variance concern:
> 1. Extreme high variance outputs inevitably **deviate from GT**, leading to **multiple negative reward signals** in accuracy based reward, that teaches the model this policy is unfeasible.
> 2. The KL divergence penalty ($-\beta \cdot \mathbb{D}_{KL}$) will **suppresses extreme outputs deviates too significantly from reference model**.
> 3. Our **normalization of output coordinate values (in 1000 x 1000 space)** and **hyperparameter $\alpha$, $\gamma$**, are designed to restrict the diversity reward from increasing blindly.
>
> For formal analysis:
> Referring to Eq (30) in [1], our diversity reward acts as an **entropy maximization reward** operating within the accuracy reward. Because both rewards are evaluated on the same set of model outputs, the **correctness constraint for diversity is implicitly inherited through the accuracy reward**.
>
> [1] Function Optimization using Connectionist Reinforcement Learning Algorithms. 1991.
> [2] Asynchronous methods for deep reinforcement learning. ICML 2016.
>
> **Q3 Q10: Literature Review: lack common CL methods**
> We leverage data reply and parameter regularization as common CL methods, evaluate them on SSv1 and SSv2 as the representative.
> **Specific results refer to rebuttal in Reviewer 4pST’s Q2 Q5.**
>
> **Q4: Mathematical Flaws in Gaussian Modeling**
> According to proof in [3, 4], the **non-diagonal elements** of the two-dimensional Gaussian covariance matrix corresponding to a rotating bounding box are equal to **$\frac{w-h}{2} \cos\theta \sin\theta$**, where $\theta$ is **rotation angle**, $w$ and $h$ are its width and height. Since UI elements’ bounding boxes (AABB) **parallel to coordinate axis**, with a **rotation angle $\theta$ is 0**, this results in **$\frac{w-h}{2} \cos\theta \sin\theta$ being 0**. Therefore, this diagonalization only reflects **UI elements have not tilted in space**, resulting in no spatial covariance on the X and Y axes, and does not hinder the ratio correlation of $w$ and $h$ themselves.
> [3] Rethinking rotated object detection with gaussian wasserstein distance loss. ICML 2021.
> [4] Learning high-precision bounding box for rotated object detection via kullback-leibler divergence. NeurIPS 2021.
>
> **Q5: Hyperparameter Sensitivity**
> Indeed, we have implemented the normalization strategy. Since we observing accuracy-based reward being in the range of [0, 3], we set $\alpha$=15 and $\gamma$=0.5 to multiple APR-iF and ARR-iF, to **ensure accuracy-based reward and diversity reward are of the same order of magnitude.** Actually, the hyperparameter sensitivity analysis is in **Figure 3**, we explore the impact of multiplying coefficients to scale hyperparameters.
>
> **Q6: Marginal Performance Gains**
> The high-resolution SSPro benchmark itself is difficult, a 2.3% improvement is a meaningful gain. Moreover, the effectiveness is clearly demonstrated in domain continual, where GUI-AiF achieves a significant performance margin over SeeClick.
>
> **Q7: N predictions used for advantage estimation**
> The default temperature setting is **0.9** to generate different sampling results for RFT.
>
> **Q8: How is the reference model $\pi_{ref}$ defined?**
> The frozen model is the one at the end of the **previous task**, to follow common continual learning setting.
>
> **Q9: Why was the Bhattacharyya distance chosen?**
> Compared to IoU, Gaussian modeling based on the Bhattacharyya distance is globally smooth, providing a **continuous reward signal for RL.** Conversely, IoU suffers from vanishing gradients: once two bounding boxes do not overlap, the IoU drops to strictly zero, causing the model to lose its optimization direction. The reward “degeneration’ explanation refers to **Q2.**
>
> **Q11: Limitations.**
> We provide 7B-scaled experiment and importance experiment of ARR-iF in **Reviewer 4pST’s Q3, Q6**, respectively.

---

> > ### Author Rebuttal · Reviewer_62uh · 2026-04-01
> >
> > I would like to thank the authors for their highly detailed and effective rebuttal. The responses have directly and convincingly addressed my core concerns.
> >
> > I am happy to raise my score to reflect the strength of this rebuttal and support its acceptance.

---

> > > ### Author Response · Authors · 2026-04-02
> > >
> > > We greatly appreciate your support and the rise in score.

---

### Official Review · Reviewer_4pST · 2026-03-12

**Soundness:** 3
**Presentation:** 3
**Significance:** 3
**Originality:** 3
**Overall Recommendation:** 4
**Confidence:** 3

**Summary:**

This paper introduces the novel task of Continual GUI Agents, addressing performance degradation of GUI agents under domain shifts (e.g., mobile to web) and resolution changes (e.g., 1080p to 4K). To solve this, the authors propose GUI-AiF (GUI-Anchoring in Flux) , a reinforcement fine-tuning framework that prevents over-adaptation to static layouts. It introduces two key rewards: APR-iF, which encourages diversity in predicted click points, and ARR-iF, which promotes diversity in predicted element regions via Bhattacharyya distance. Experiments show significant gains over SFT and RFT baselines on ScreenSpot benchmarks, establishing a foundation for continual learning in GUI agents.

**Compliance With Llm Reviewing Policy:**

Affirmed.

**Final Justification:**

The rebuttal successfully addresses all my prior concerns. I have therefore selected “Fully resolved” and raised my score.

**Key Questions For Authors:**

1. Several recent works (e.g., GUI-R1, UI-R1, InfiGUI-R1) already use reinforcement learning to improve GUI grounding. How does GUI-AiF differ from these approaches beyond reward formulation? Could the proposed rewards be directly integrated into these frameworks?
2. The paper mainly compares against standard GUI grounding models. It would be useful to compare with general continual learning methods (e.g., replay-based or regularization-based methods) applied to GUI agents.
3. The framework introduces APR-iF and ARR-iF. Could the authors provide a clearer ablation showing whether both rewards are necessary? For example, does APR-iF alone already achieve most of the improvement?

**Limitations:**

yes

**Strengths And Weaknesses:**

Strengths:
1. Most existing GUI agent works focus on improving grounding accuracy in static datasets (e.g., SeeClick, GUI-R1, InfiGUI-R1). This paper highlights the realistic scenario where GUI environments change over time, such as switching from mobile apps to web interfaces or moving from 1080p to 4K resolution.
2. Clear formulation of continual GUI learning scenarios. The paper defines two concrete settings: domain-in-flux and resolution-in-flux, which are intuitive and relevant for real-world GUI automation.
3. Simple reward-based design. The proposed rewards (APR-iF and ARR-iF) aim to encourage exploration of interaction points and diversity of predicted regions, which may help improve robustness when UI layouts or scales change.

Weaknesses
1. Incremental novelty relative to existing GUI RL methods. Recent works such as GUI-R1, UI-R1, and Visual-RFT also use reinforcement-based reward modeling to improve GUI grounding. Compared with these works, the proposed contribution mainly lies in adapting reward design for continual settings rather than introducing a fundamentally new learning framework.
2. Limited exploration of broader continual learning techniques. The method focuses on reward shaping but does not explore other common continual learning strategies (e.g., replay buffers, parameter regularization, or modular architectures). It is unclear whether these alternatives would perform similarly or better.
3. Experimental scope is relatively limited. The experiments mainly use a 3B-scale model and a limited number of datasets and benchmarks, which may restrict conclusions about scalability and generalization.

---

> ### Author Rebuttal · Authors · 2026-03-31
>
> **Q1 Q4: Novelty relative to existing GUI RL methods.**
> Visual-RFT leverages verifiable rewards (e.g., IoU) and GRPO to improve models’ visual task (including grounding); GUI-R1 and UI-R1 adopt an interaction point reward to improve GUI grounding performance; InfiGUI-R1 explores the IoU reward between predicted region and GT’s region to enhance GUI grounding.
> **All of these methods focus on designing reward policies to improve performance on fixed training data, are not suitable for the scenario where GUI agents are required to adapt to dynamic UI environment.** We address this by proposing GUI-AiF that introduces diversity rewards on UI interaction points and regions. GUI-AiF acts as an **entropy maximization policy within RL (refer to Reviewer jNMm’s [1, 2]).** This high entropy policy can **enhance model's exploration capability**. Based on this advantage, GUI-AiF help GUI agents **better adapt to different UI interfaces.** Beside, GUI-AiF **can be integrated into any RFT-based GUI method** that formulates point and region, including the mentioned GUI RL method GUI-R1, UI-R1 and InfiGUI-R1. As discussed above, GUI-AiF differs fundamentally from prior work in principles, scope, and objectives.
> [1] Function Optimization using Connectionist Reinforcement Learning Algorithms. 1991.
> [2] Asynchronous methods for deep reinforcement learning. ICML 2016.
>
> **Q2 Q5: Limited exploration of CL techniques.**
> We leverage **data reply** and **parameter regularization** as common CL methods, evaluate them on SSv1 and SSv2 as the representative.
> We first conduct data reply by adding part of data from previous tasks into new task.
> SSv1:
>
> |Method|Setting|Mobile| |Desktop| |Web| |Overall|
> |:---|:---|:---|:---|:---|:---|:---|:---|:---|
> | | |Text|Icon|Text|Icon|Text|Icon|Avg.|
> |Data-Reply|M.→D.→W.|93.4|72.1|87.1|61.4|81.3|58.3|75.6|
> |GUI-AiF|M.→D.→W.|**96.1**|**76.9**|**95.7**|**68.3**|**84.8**|**68.2**|**81.7**|
> SSv2:
>
> |Method|Setting|Mobile| |Desktop| |Web| |Overall|
> |:---|:---|:---|:---|:---|:---|:---|:---|:---|
> | | |Text|Icon|Text|Icon|Text|Icon|Avg.|
> |Data-Reply|M.→D.→W.|94.8|76.3|86.6|67.9|82.5|61.6|78.3|
> |GUI-AiF|M.→D.→W.|**96.6**|**83.6**|**90.2**|**77.4**|**82.9**|**70.5**|**83.5**|
>
> Then we choose **LwF [3] as parameter regularization**, and build a baseline for SFT+LwF by measuring differences between training model and base model using KL terms (RFT naturally has KL items, so this method only can be integrated into SFT).
> [3] Learning without Forgetting, 2018
> SSv1:
>
> |Method|Setting|Mobile| |Desktop| |Web| |Overall|
> |:---|:---|:---|:---|:---|:---|:---|:---|:---|
> | | |Text|Icon|Text|Icon|Text|Icon|Avg.|
> |SFT+LwF|M.→D.→W.|92.3|75.1|86.1|55.7|77.8|55.3|73.7|
> |GUI-AiF|M.→D.→W.|**96.1**|**76.9**|**95.7**|**68.3**|**84.8**|**68.2**|**81.7**|
> SSv2:
>
> |Method|Setting|Mobile| |Desktop| |Web| |Overall|
> |:---|:---|:---|:---|:---|:---|:---|:---|:---|
> | | |Text|Icon|Text|Icon|Text|Icon|Avg.|
> |SFT+LwF|M.→D.→W.|91.7|78.2|84.0|60.7|76.5|55.2|74.4|
> |GUI-AiF|M.→D.→W.|**96.6**|**83.6**|**90.2**|**77.4**|**82.9**|**70.5**|**83.5**|
>
> The results indicate that common CL methods are unable to achieve effective continual learning performance for GUI agents.
>
> **Q3: Experimental scope is relatively limited.**
> We provide a set of results based on **Qwen2.5-VL-7B** under SSv2 and SSPro benchmarks. We can find that larger scaled model further improve the performance.
> SSv2:
>
> |Method|Setting|Mobile| |Desktop| |Web| |Overall|
> |:---|:---|:---|:---|:---|:---|:---|:---|:---|
> | | |Text|Icon|Text|Icon|Text|Icon|Avg.|
> |GUI-AiF (3B)|M.→D.→W.|**98.1**|85.3|92.7|80.2|81.2|74.9|85.4|
> |GUI-AiF (7B)|M.→D.→W.|98.0|**90.1**|**94.5**|**82.3**|**86.2**|**78.5**|**88.3**|
>
> SSPro:
>
> |Method|Setting|CAD| |Dev| |Creative| |Scientific| |Office| |OS| | |
> |:---|:---|:---|:---|:---|:---|:---|:---|:---|:---|:---|:---|:---|:---|:---|
> | | |Text|Icon|Text|Icon|Text|Icon|Text|Icon|Text|Icon|Text|Icon|Avg.|
> |GUI-AiF (3B)|N.→H.|20.3|15.6|29.2|2.1|23.2|**3.5**|33.3|14.7|38.4|**18.2**|22.4|6.8|19.0|
> |GUI-AiF (7B)|N.→H.|**25.1**|**18.9**|**33.5**|**7.8**|**26.7**|3.3|**37.9**|**19.2**|**43.7**|17.9|**26.7**|**8.9**|**22.5**|
>
> We specifically select these benchmarks that comply with the continuous GUI agent to train and evaluate.
>
> **Q6: APR-iF and ARR-iF ablation show whether both rewards are necessary? Does APR-iF alone already achieve most of the improvement?**
> ARR-iF’s importance is **more evident in benchmark with significant differences in element box scale.** To verify this, we select samples with significant differences in element box scale from mobile, desktop, and web subsets of SSv2. The results are as follows:
>
> |Method|Setting|Mobile| |Desktop| |Web| |Overall|
> |:---|:---|:---|:---|:---|:---|:---|:---|:---|
> | | |Text|Icon|Text|Icon|Text|Icon|Avg.|
> |GUI-AiF|M.→D.→W.|98.1|85.3|92.7|80.2|81.2|74.9|85.4|
> |APR-iF| M.→D.→W. |93.2|83.3|90.8|70.0|80.6|68.7|81.1|
> |ARR-iF| M.→D.→W. |95.7|80.1|91.3|73.5|79.7|64.5|80.8|

---

> > ### Author Rebuttal · Reviewer_4pST · 2026-04-02
> >
> > All my original concerns have been resolved.

---

> > > ### Author Response · Authors · 2026-04-02
> > >
> > > We greatly appreciate your rise in score.

---

### Official Review · Reviewer_jNMm · 2026-03-12

**Soundness:** 4
**Presentation:** 2
**Significance:** 3
**Originality:** 3
**Overall Recommendation:** 4
**Confidence:** 5

**Summary:**

This paper introduces Continual GUI Agents, a new task setting where GUI agents must adapt to evolving graphical interfaces as domains (e.g., mobile, desktop, web) and screen resolutions change over time. The authors argue that existing GUI grounding methods trained on static datasets struggle under such distribution shifts. To address this, they propose GUI-AiF, a reinforcement fine-tuning framework that encourages robust grounding through two reward components: Anchoring Point Reward in Flux (APR-iF), which promotes diversity in predicted interaction points, and Anchoring Region Reward in Flux (ARR-iF), which encourages adaptability to varying element scales. Experiments on different benchmarks demonstrate improved continual learning performance compared to existing baselines. Overall, this study's significant contribution comprises introducing the continual GUI agent setting and a reward-driven reinforcement framework for encouraging exploration under shifting environments.

**Compliance With Llm Reviewing Policy:**

Affirmed.

**Final Justification:**

The rebuttal resolved my main concerns by explicitly connecting the method to entropy-maximization objectives, adding the requested domain-agnostic EMR baseline, clarifying the train/eval protocol, and answering the reward-design questions around normalization and where the auxiliary reward is applied. I am therefore maintaining my score of 4: the paper makes a solid contribution by introducing the continual GUI-agent setting and showing clear empirical gains, though the domain-aware reward formulation still feels somewhat incremental, which is why I am not scoring it higher.

**Key Questions For Authors:**

1. Are screen coordinates normalized across resolutions when computing the rewards? I could see changes in resolution causing some variance issues in the rewards since they directly depend on screen coordinates.
2. Have you tried adding APR-iF/ARR-iF rewards directly to the base reward scores, instead of adding them to the advantages (eq 6)? Adding rewards post-normalization increases variance with no clear benefit so I'm surprised of this design choice.
3. There are two GUI-AiF rows in table 2, what is the difference between the two?

**Limitations:**

yes

**Strengths And Weaknesses:**

**Strenghts**

The paper addresses an important and under-explored setting for GUI agents: continual learning under evolving interfaces. Evaluating agents under domain and resolution shifts better reflects real-world deployment scenarios, where GUI layouts and screen resolutions change over time rather than remaining static.

The method proposed is simple, low-compute and easy to implement. The reward shaping strategy integrates naturally into existing reinforcement fine-tuning pipelines and should be straightforward to implement and reproduce.

The experimental evaluation is thorough and clearly presented. The authors compare against multiple supervised fine-tuning and reinforcement fine-tuning baselines across several benchmarks and continual-learning scenarios.

The proposed approach achieves higher task success rate than both SFT and RFT baselines, suggesting that the proposed rewards improve learning within shifting GUI environments.

**Weaknesses**

**Missing references in related work.** The reward functions proposed are essentially domain-specific entropy maximization rewards over the agent's policy. The aim of this class of objective is to encourage exploration by keeping the entropy over actions high. Entropy maximization rewards have been around in RL for a long time [1] and are still relevant in deep RL [2]. This work would clearly benefit from making this connection apparent.

**Too few baselines.** The authors should include at least one domain-agnostic entropy maximization reward function as a baseline to demonstrate the usefulness of their domain-specific reward function design for GUI domains. A possible domain-agnostic entropy maximization reward function could be to maximize variance over the "raw actions" outputted by the policy. As I understand these raw actions would be the numerical coordinates for the bounding box corners.

Similarly there were no Continual RL methods in the related work or used as baselines, which I found surprising given the topic of the paper.

**Evaluation set construction.** Information is currently missing on the evaluation set construction. Is there a proper train/eval separation? How many / what percentage of each dataset is reserved as evaluation tasks?

**Improvements needed on formatting and presentation:**

I liked the inclusion of pictograms for the APR-iF and ARR-iF reward functions, I suggest to include them everywhere in the main text as the APR/ARR acronyms look too visually similar otherwise.

While the paper is relatively easy to follow overall, there are several sections that need improvements to formulation/clarity. I'll include spots that need editing below:
- l90-94 rhs: sentence isn't grammatically correct / hard to follow.
- l141 rhs: remove "this"
- l148 lhs: a -> an
- l306 lhs: adopted -> adapted
- l326 rhs: sentence isn't grammatically correct / hard to follow.
- l365 lhs: sentence isn't grammatically correct / hard to follow.
- l357 rhs. Divergency -> Divergence
- I couldn't follow what section A of appendix was trying to say.


[1] WILLIAMS, R. J., & PENG, J. (1991). Function Optimization using Connectionist Reinforcement Learning Algorithms. Connection Science, 3(3), 241–268. https://doi.org/10.1080/09540099108946587

[2] Mnih, V., Badia, A. P., Mirza, M., Graves, A., Lillicrap, T., Harley, T., ... & Kavukcuoglu, K. (2016, June). Asynchronous methods for deep reinforcement learning. In International conference on machine learning (pp. 1928-1937). PmLR.

---

> ### Author Rebuttal · Authors · 2026-03-31
>
> **Q1: Missing references in related work.**
> Our reward function (APR-iF and ARR-iF) encourages relatively **high entropy** over the predicted **interaction points** and **regions**. As you noted, this reward essentially aligns GUI agent behaviors with the principle of **maximum entropy** [1,2,3,4]. We appreciate your insight and have highlighted this connection in the revised version to clarify its theoretical significance.
> [1] GTPO and GRPO-S: Token and sequence-level reward shaping with policy entropy. 2025.
> [2] Arbitrary entropy policy optimization: Entropy is controllable in reinforcement finetuning. 2025.
> [3] Function Optimization using Connectionist Reinforcement Learning Algorithms. 1991.
> [4] Asynchronous methods for deep reinforcement learning. ICML 2016.
>
> **Q2: Too few baselines.**
> Following your suggestion, we maximize variance over the raw actions as a **Entropy Maximization Reward** (EMR) baseline. In detail, we compute the corresponding coordinate variance across N predicted bounding boxes [x_1, y_1, x_2, y_2]. We evaluated EMR on the SSv1 and SSv2, and results are:
> SSv1:
>
> |Method|Setting|Mobile| |Desktop| |Web| |Overall|
> | :--- | :--- | :--- | :--- | :--- | :--- | :--- | :--- | :--- |
> | | | Text | Icon | Text | Icon | Text | Icon | Avg. |
> | EMR | M.→D.→W. | 94.2 | 76.8 | 94.3 | 62.5 | 77.9 | 52.8 | 76.4 |
> | GUI-AiF | M.→D.→W. | **96.1** | **76.9** | **95.7** | **68.3** | **84.8** | **68.2** | **81.7** |
> SSv2:
>
> | Method | Setting | Mobile | | Desktop | | Web | | Overall |
> | :--- | :--- | :--- | :--- | :--- | :--- | :--- | :--- | :--- |
> | | | Text | Icon | Text | Icon | Text | Icon | Avg. |
> | EMR | M.→D.→W. | 92.3 | 80.2 | 88.5 | 76.3 | 79.8 | 68.7 | 80.9 |
> | GUI-AiF | M.→D.→W. | **96.6** | **83.6** | **90.2** | **77.4** | **82.9** | **70.5** | **83.5** |
>
> This EMR yielded poor continual learning results, as computing raw coordinate variance only alters the box's scale and ratio without capturing the physical features of GUI elements. We will discuss continual RL methods in related work.
>
> **Q3: Evaluation set construction.**
> We maintain consistency with mainstream GUI agents [5, 6, 7], which separately train and evaluate on different GUI benchmarks. We will further clarify this evaluation setting in the revised manuscript.
> [5] Infigui-g1: Advancing gui grounding with adaptive exploration policy optimization. AAAI 2026
> [6] GUI-G^2: Gaussian Reward Modeling for GUI Grounding." AAAI 2026
> [7] UI-Ins: Enhancing GUI Grounding with Multi-Perspective Instruction-as-Reasoning. ICLR 2026
>
> **Q4: Improvements needed on formatting and presentation.**
> Thanks for your corrections, we will polish our manuscript.
>
> **Q5: Are screen coordinates normalized across resolutions when computing the rewards?**
> We confirm that our reward calculation is unaffected by resolution. To achieve this, we normalize all interface element coordinates to a **1000 x 1000** space prior to the reward computation.
>
> **Q6: Adding APR-iF/ARR-iF rewards directly to base reward scores. Adding rewards post-normalization increases variance with no clear benefit.**
> In GRPO, the advantage $\mathrm{A_i}$ is computed by normalizing the raw rewards $r_i$ across $N$ outputs: $A_i = \frac{r_i - \mu}{\sigma}$ (shown in eq 5). Crucially, our proposed GUI-AiF reward is to calculate a value from a set of $N$ outputs, where the value is an identical constant $C$. **If we follow the suggestion of adding it to the base reward** prior to normalization (i.e., $r_i'=r_i+C$), the new group mean would become $μ'=μ+C$. During the advantage normalization step, this **GUI-AiF reward would be mathematically cancelled out entirely**: $A_i' = \frac{(r_i + C) - (\mu + C)}{\sigma} = \frac{r_i - \mu}{\sigma} = A_i.$. This implies that if added prior to normalization, the gradient associated with GUI-AiF reward would be zero, rendering the agent incapable of learning the exploration strategy. Therefore, adding GUI-AiF reward after $\mathrm{A_i}$ is to encourage the exploration behavior of model.
>
> **Q7 There are two GUI-AiF rows in Table 2. What is the difference between the two?**
> **These two GUI-AiF represent the reward integrated into different baselines**. By demonstrating the improvements on both a baseline (using Euclidean/IoU modeling) and a probabilistic one (using Gaussian distributions), we verify the generalizability of our approach. Essentially, any RFT-based GUI agent that formulates interaction points and/or regions can use the GUI-AiF reward to enhance its continual adaptation across diverse UI environments.

---

> > ### Author Rebuttal · Reviewer_jNMm · 2026-04-03
> >
> > The authors have addressed my questions, I have no further concerns.

---

> > > ### Author Response · Authors · 2026-04-04
> > >
> > > We greatly appreciate your comments and reply.

---

### Decision · Program_Chairs · 2026-04-30

**Decision:**

Accept (regular)

**Comment:**

This paper introduces Continual GUI Agents, a novel task requiring GUI agents to adapt under domain and resolution shifts, proposing GUI-AiF with entropy-maximizing rewards (APR-iF for point diversity, ARR-iF for region diversity). All four reviewers suggest Weak Accept (score 4, confidence 3-5) with perfect consensus: every concern was marked "Fully resolved" post-rebuttal. Strengths: novel and important task, well-motivated, simple practical method, comprehensive experiments with ablations. Initial concerns (missing entropy RL literature, insufficient baselines, scope limitations, mathematical justifications) all convincingly addressed in rebuttal. Reviewers explicitly state willingness to raise scores or strongly support acceptance. Thus I recomment "ACCEPT".